# 4DGCPro: Efficient Hierarchical 4D Gaussian Compression for Progressive Volumetric Video Streaming

**Zihan Zheng[1] , Zhenlong Wu[1] , Houqiang Zhong[2] , Yuan Tian[2,3] , Ning Cao[4] ,**
**Lan Xu[5] , Jiangchao Yao[1] , Xiaoyun Zhang[1] , Qiang Hu[1]\*, Wenjun Zhang[1,2]**

Cooperative Medianet Innovation Center, Shanghai Jiaotong University[1]
Department of Electronics, Shanghai Jiaotong University[2]
Shanghai AI Lab[3]
Cloud platform department, E-surfing Vision Technology Co., Ltd.[4]
School of Information Science and Technology, ShanghaiTech University[5]

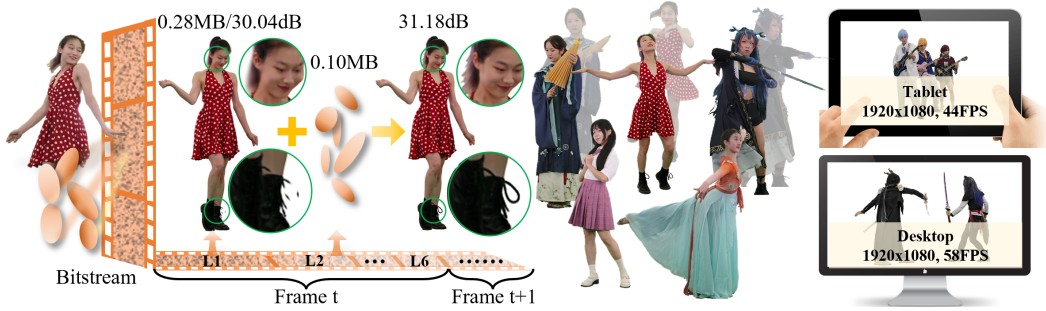

Figure 1: **Left:** Our method enables progressive streaming of hierarchical 4D Gaussians within a single bitstream, where incremental enhancement layers (e.g., +0.10MB) gradually improve visual quality (e.g., from 30.04dB to 31.18dB) with minimal bitrate overhead. **Right:** The streamed content is adaptively decoded and rendered in real-time on various devices (e.g., tablets 44FPS, desktops 58FPS) by dynamically selecting layers (L1-L6) based on available bandwidth and compute.

## Abstract

Achieving seamless viewing of high-fidelity volumetric video, comparable to 2D video experiences, remains an open challenge. Existing volumetric video compression methods either lack the flexibility to adjust quality and bitrate within a single model for efficient streaming across diverse networks and devices, or struggle with real-time decoding and rendering on lightweight mobile platforms. To address these challenges, we introduce 4DGCPro, a novel hierarchical 4D Gaussian compression framework that facilitates real-time mobile decoding and high-quality rendering via progressive volumetric video streaming in a single bitstream. Specifically, we propose a perceptually-weighted and compression-friendly hierarchical 4D Gaussian representation with motion-aware adaptive grouping to reduce temporal redundancy, preserve coherence, and enable scalable multi-level detail streaming. Furthermore, we present an end-to-end entropy-optimized training scheme, which incorporates layer-wise rate-distortion (RD) supervision and attribute-specific entropy modeling for efficient bitstream generation. Extensive experiments show that 4DGCPro enables flexible quality and multiple bitrate within a single model, achieving real-time decoding and rendering on mobile devices while outperforming existing methods in RD performance across multiple datasets.

---

\*The corresponding author is Qiang Hu(qiang.hu@sjtu.edu.cn)

39th Conference on Neural Information Processing Systems (NeurIPS 2025).

# 1 Introduction

Volumetric video enables immersive 3D experiences with free-viewpoint navigation, but streaming and rendering high-quality long sequences with large motions remains challenging, especially on lightweight devices like mobile phones. Compared to 2D video, volumetric content demands higher bandwidth, storage, and real-time decoding capabilities, making fixed-bitrate solutions inadequate to handle the variability across heterogeneous devices and network conditions. Therefore, the core challenge lies in achieving real-time, high-fidelity playback with low computational cost, while enabling scalable and progressive streaming under constrained resources.

Traditional volumetric reconstruction methods, including surface estimation [8], point clouds [57], meshes [68], light field [28, 44] and depth-based techniques [60, 79], struggle to faithfully capture the geometric complexity and temporal dynamics of real-world scenes. Neural radiance field (NeRF) [40] address these limitations by modeling view-dependent appearance without relying on explicit geometry, enabling photorealistic rendering. While extensions [49, 30, 12, 24, 14, 5, 78] adapt NeRF to dynamic scenes, they remain constrained by the difficulty of handling long sequences and efficient streaming. Some works [66, 67, 73, 81, 80] compress dynamic NeRFs to enable streaming, but the high computational cost of decoding and rendering limits their practicality in real-time applications.

Recent work on 3D Gaussian Splatting (3DGS) [27] introduces an explicit scene representation using anisotropic Gaussian primitives with real-time, differentiable rasterization, achieving unprecedented rendering speed and visual quality. Subsequent studies [34, 72, 19, 75] extend 3DGS to dynamic scenes by incorporating temporal attributes into Gaussian parameters, but require full-sequence pre-loading during training and rendering, limiting streaming practicality. Alternative approaches [38, 61, 18, 16] model temporal Gaussian variations via deformable fields or residual tracking to enable streamable representations, yet incur substantial bandwidth overhead. A few studies [20, 26, 25, 69] have explored compression for dynamic 3DGS. For example, 4DGC [20] jointly optimizes representation and entropy models via RD loss, improving efficiency but struggling with high decoding latency and poor robustness to large motions due to rigid modeling. More fundamentally, existing dynamic Gaussian compression methods lack the flexibility to adjust video quality and bitrate within a single model, and typically require separate models for each bitrate, leading to high storage costs and limited adaptability under varying network and device conditions.

To tackle the above challenges, we propose 4DGCPro, a novel hierarchical 4D Gaussian compression approach for progressive volumetric video streaming. As illustrated in Fig. 1, our method achieves multiple bitrate using a single model and enables real-time decoding and high-fidelity rendering on lightweight devices for large-motion sequences. We realize this through three key innovations. First, we introduce a perceptually-weighted hierarchical Gaussian representation for keyframes, guided by a significance metric that combines geometric volume and opacity. This enables scalable representation across detail levels and establishes the foundation for dynamic modeling. Second, we propose a hierarchical motion modeling strategy, where motion in subsequent frames is decomposed into rigid transformations and residual deformations to capture large displacements and preserve temporal coherence. We further adopt motion-aware adaptive Gaussian grouping to handle topological changes and long-term dynamics, ensuring compact and consistent temporal representation.

Third, we propose a joint entropy-optimized training and progressive coding framework for efficient and scalable bitstream generation. Specifically, we introduce layer-wise rate-distortion (RD) supervision into the training pipeline using differentiable quantization and attribute-specific entropy modeling. For keyframes, we utilize FFT-accelerated Gaussian kernel density estimation (KDE) for precise bitrate prediction of Gaussian attributes, with hierarchical Gaussian optimization guided by per-layer RD trade-offs. For inter-frame, we apply Gaussian-distribution-based entropy estimation and temporal consistency constraints to maintain compactness and coherence. After training, we quantize attributes and convert multi-layer representations into stacked 2D single-channel maps, which are encoded with 2D codecs into a progressive bitstream, enabling scalable real-time decoding and rendering via hardware video codecs and shaders. Experimental results show that our 4DGCPro supports multiple bitrates using a single model and achieves state-of-the-art RD performance across various datasets. Compared to the SOTA method HPC [80], our approach achieves a **3x** compression rate without quality degradation, while enabling real-time decoding and rendering on mobile devices.

In summary, our contributions are as follows:

- We propose 4DGCPro, a novel framework for progressive volumetric video streaming that supports multiple bitrates with a single compact model, enabling real-time decoding and rendering on mobile devices with superior RD performance.
- We introduce a compact hierarchical 4D Gaussian representation with motion-aware adaptive grouping for scalable and high-fidelity modeling of dynamic scenes.
- We present an end-to-end entropy-optimized training scheme with layer-wise RD supervision and attribute-specific entropy modeling, enabling fine-grained RD optimization across layers and better overall compression.

## 2 Related Work

### 2.1 NeRF-based Volumetric Video Modeling

NeRF [40] have revolutionized 3D scene representation using differentiable volume rendering with implicit neural representations. While recent advances in static scene representation [2–4, 7, 39, 41, 46, 52, 53] have improved compactness and reconstruction speed, several works have extended these methods to dynamic scenes. Flow-based approaches [32, 33] construct 3D features from monocular video, reducing data collection complexity but requiring additional priors for complex scenes. Deformation field methods [10, 45, 49, 59] warp dynamic frames into a canonical space to capture temporal features, yet suffer from slow training and rendering. To accelerate performance, recent methods [12, 24, 14, 5, 58, 31, 47, 63, 65] adopt explicit 4D radiance field representations based on structured volumetric decompositions (e.g., voxel grids, multi-plane projections, or tensor factorizations), yet these unified frameworks remain incompatible with streaming scenarios.

### 2.2 3DGS-based Volumetric Video Modeling

3DGS [27] and its variants [22, 13, 23, 6, 17]enable photorealistic static scene reconstruction through their efficiency and physical interpretability, with recent extensions to dynamic scenes. Current dynamic 3DGS approaches mainly follow two paradigms. Some studies [34, 72, 76, 19, 75, 74, 77] model Gaussian attributes as temporal functions to create unified dynamic representations, which achieve exceptional RD performance but neglect streaming feasibility. Alternative approaches [38, 61, 18, 16] employ frame-wise modeling with explicit rigid motion estimation, enabling streamable Gaussian volumetric video at the cost of increased data volume and compromised reconstruction quality. While $V^3$ [69] optimizes the full Gaussian coefficients to model complex motions, its fixed group length leads to error accumulation or data redundancy. Meanwhile, it lacks the capability to support multiple-bitrate selection within a single bitstream. Our approach introduces a compact hierarchical motion-aware Gaussian representation coupled with adaptive Gaussian grouping that dynamically responds to topological changes and achieves multiple bitrate using a single model.

### 2.3 Volumetric Video Compression

Volumetric video compression is crucial for reducing massive data requirements, where traditional approaches employ octree [55, 62] and wavelet [42] techniques (later standardized as MPEG-PCC [56]), while subsequent learning-based methods [35, 50, 51, 29, 1, 15, 64] focus on improved efficiency. While recent advances [66, 67, 73, 59, 48, 9, 54, 81, 80, 21] have made progress in compressing dynamic NeRF features for storage optimization, they commonly suffer from poor quality and slow decoding/rendering. For instance, HPC [80] employs learned compression for progressive coding of residual feature grids representing dynamic scenes. However, its high decoding latency limits real-time applications. For 3DGS-based methods, static scene techniques [43, 11, 37, 71] dominate, whereas dynamic scene approaches [26, 25, 20] face inefficiency and single-rate constraints. Our method delivers superior RD performance and computational efficiency using standard video codecs, supporting both hardware-accelerated real-time decoding and progressive streaming for quality adaptation across dynamic bandwidth.

## 3 Method

In this section, we present the technical details of our 4DGCPro architecture (Fig. 2). The framework begins with a perceptually-weighted hierarchical Gaussian representation for keyframes (Sec. 3.1),

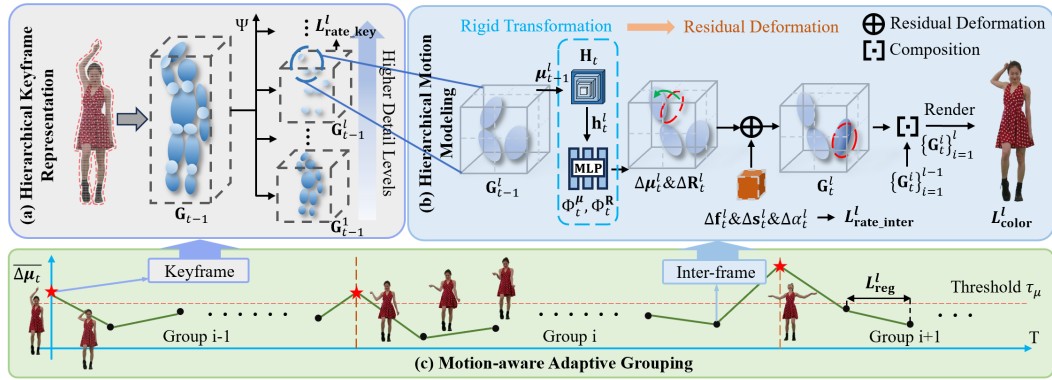

Figure 2: **Our 4DGCPro framework.** (**a**) Perceptually-weighted hierarchical 4D Gaussian representation models keyframes at multi-level detail for progressive reconstruction. (**b**) Hierarchical motion modeling decomposes dynamic scenes into rigid transformations and residual deformations based on the previous frame, while (**c**) motion-aware adaptive grouping dynamically adjusts to topological changes to enhance temporal consistency and reduce error accumulation. The entire pipeline is end-to-end optimized with layer-wise RD supervision and attribute-specific entropy modeling.

which establishes the foundation for dynamic scene characterization and progressive streaming. We then introduce a hierarchical motion modeling approach with adaptive grouping (Sec. 3.2), decomposing motions into rigid transformations and residual deformations. This motion-aware adaptive Gaussian grouping mechanism effectively handles diverse motion patterns in complex scenes. To generate efficient and scalable bitstreams, we incorporate layer-wise RD optimization into the training pipeline through differentiable quantization and attribute-specific entropy modeling, followed by compression using standard 2D video codecs (Sec. 3.3).

### 3.1 Perceptually-Weighted Hierarchical Gaussian Keyframe Representation

Recall that 3DGS represents scenes explicitly through 3D Gaussians $\mathbf{G}$, defined by a set of learnable parameters, including center position $\boldsymbol{\mu}$, rotation matrix $\mathbf{R}$ representing orientation, spherical harmonic coefficients $\mathbf{f}$ for view-dependent appearance modeling, scaling factors $\mathbf{s}$ controlling spatial extent, and opacity value $\alpha$. The spatial influence at point $\mathbf{x}$ follows $\mathbf{G}(\mathbf{x})$, expressed as:

$$\mathbf{G}(\mathbf{x}) = \exp\left(-\frac{1}{2}(\mathbf{x} - \boldsymbol{\mu})^T \boldsymbol{\Sigma}^{-1}(\mathbf{x} - \boldsymbol{\mu})\right). \tag{1}$$

The covariance matrix $\boldsymbol{\Sigma}$ is constructed through $\boldsymbol{\Sigma} = \mathbf{R}\mathbf{s}\mathbf{s}^T\mathbf{R}^T$. With $\alpha'_i$ being the projection of the opacity of the $i$-th Gaussian onto the image plane and $\mathbf{c}_i$ denoting the color of the $i$-th Gaussian in the viewing direction, the pixel color $\mathbf{c}$ is computed by differentiable splatting of $N$ ordered Gaussians as follows:

$$\mathbf{c} = \sum_{i \in N} \mathbf{c}_i \alpha'_i \prod_{j=1}^{i-1}(1 - \alpha'_j), \tag{2}$$

When reconstructing long-duration dynamic scenes, we first reconstruct high-quality keyframes to serve as references for subsequent inter-frames. Inspired by V$^3$ [69], we initialize keyframe 3D Gaussians through NeuS2-based [70] surface mesh extraction. After pre-training, low-opacity Gaussians are pruned to achieve compact representations. While this optimized 3DGS delivers high-fidelity reconstruction, its substantial data footprint becomes problematic for smooth viewing under fluctuating bandwidth conditions. We therefore propose a perceptually-weighted hierarchical Gaussian representation guided by significance metric $\Psi$, which serves as the basis for progressive transmission and rendering. The proposed metric $\Psi$ analytically evaluates each Gaussian's visual importance through two geometrically-grounded attributes: (1) spatial volume $S$, representing the 3D volume occupied by the Gaussian and computed as $\frac{4}{3}\pi abc$ (where $a$, $b$, $c$ are its scale parameters along the three principal axes), which reflects its structural contribution to the scene geometry; and (2) opacity $\alpha$, which determines its perceptual weight in final rendering. These orthogonal factors are integrated with the weight $\lambda_\Psi$:

$$\Psi = \alpha + \lambda_\Psi S. \tag{3}$$

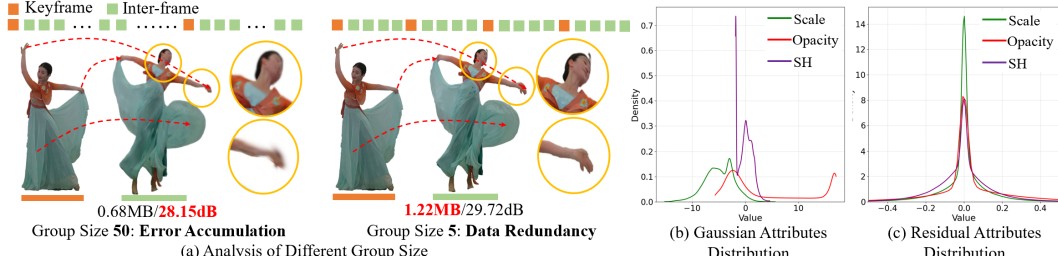

Figure 3: Analysis of group size and attributes distributions. **(a)** Large groups suffer from error accumulation while small groups exhibit data redundancy. **(b)** Keyframe Gaussian attributes display irregular spatial distributions, whereas **(c)** residual attributes follow Gaussian distributions.

After sorting all Gaussians in descending order using our significance metric, we partition them into $L$ hierarchical layers $\mathbf{G} = \{\mathbf{G}^l\}_{l=1}^{L}$. The base layer $\mathbf{G}^1$ preserves essential scene structures, while subsequent layers progressively enhance details. This hierarchical representation facilitates adaptive streaming, where the client dynamically selects the optimal number of layers $l$ to decode based on network conditions and computational resources. This approach ensures smooth playback while efficiently balancing transmission overhead and reconstruction fidelity across diverse network environments.

**Progressive Rendering.** Our method supports progressive rendering from a single compressed representation, enabling scalable visual output with adjustable levels of detail. Starting from the base layer ($l = 1$), which contains essential structural and appearance information, each subsequent Gaussian layer incrementally refines the reconstruction. Specifically, when decoding up to level $l$, only the Gaussians up to that layer (denoted by the index set $N^l$) are used for rendering. The color $\mathbf{c}^l$ at this stage is computed as:

$$\mathbf{c}^l = \sum_{i \in N^l} \mathbf{c}^i {\alpha'}_i \prod_{j=1}^{i-1}(1 - {\alpha'}_j). \tag{4}$$

Each additional layer introduces a compact set of Gaussians that enhance detail without redundant transmission. This hierarchical refinement strategy allows the model to adapt dynamically to available computational and bandwidth resources, balancing reconstruction quality and efficiency in real time. The result is a scalable, high-fidelity rendering system capable of maintaining seamless visual enhancement under varying resource constraints, all within a unified bitstream.

### 3.2 Hierarchical Motion Modeling with Adaptive Grouping

Building upon the hierarchical keyframe Gaussian representation, we employ it as a reference basis for training subsequent inter-frames. Our proposed hierarchical motion modeling strategy effectively captures large-scale complex motions while maintaining temporal coherence through decomposition of frame differences into rigid transformations and residual deformations. We futher introduces motion-aware adaptive Gaussian grouping, which dynamically responds to varying scene changes to achieve dual benefits: enhanced representation fidelity and reduced model size for efficient streaming.

**Rigid Transformation.** To estimate the rigid transformations of Gaussians between frames, we utilize the Gaussian positions $\boldsymbol{\mu}_{t-1} = \{\boldsymbol{\mu}_{t-1}^l\}_{l=1}^{L}$ from the previous frame as input and predicts both translation $\Delta\boldsymbol{\mu}_t = \{\Delta\boldsymbol{\mu}_t^l\}_{l=1}^{L}$ and rotation $\Delta\mathbf{R}_t = \{\Delta\mathbf{R}_t^l\}_{l=1}^{L}$. The module first employs a multi-resolution hash grid $\mathbf{H}_t = \{\mathbf{H}_t^l\}_{l=1}^{L_h}$ with $L_h$ levels to capture motion features $\mathbf{h}_t$ at different scales through hash coding as:

$$\mathbf{h}_t^l = \{\mathbf{h}_t^{l_h}\}_{l_h=1}^{L_h} = \{\text{interp}(\boldsymbol{\mu}_{t-1}^l, \mathbf{H}_t^{l_h})\}_{l_h=1}^{L_h}, \tag{5}$$

where interp($\cdot$) refers to the hash grid interpolation operation. Subsequently, $\mathbf{h}_t$ is input into two lightweight MLPs, namely $\Phi_t^{\boldsymbol{\mu}}$ and $\Phi_t^{\mathbf{R}}$, to calculate the translation $\Delta\boldsymbol{\mu}_t^l$ and rotation $\Delta\mathbf{R}_t^l$ for each Gaussian:

$$\Delta\boldsymbol{\mu}_t^l = \Phi_t^{\boldsymbol{\mu}}(\mathbf{h}_t^l), \quad \Delta\mathbf{R}_t^l = \Phi_t^{\mathbf{R}}(\mathbf{h}_t^l). \tag{6}$$

In this manner, the position and rotation of frame $t$ can be determined using the equations $\boldsymbol{\mu}_t^l = \boldsymbol{\mu}_{t-1}^l + \Delta\boldsymbol{\mu}_t^l$ and $\mathbf{R}_t^l = \Delta\mathbf{R}_t^l\mathbf{R}_{t-1}^l$.

**Residual Deformation.** Existing motion-aware 3DGS streaming methods [20, 61] primarily focus on rigid motion simulation and Gaussian compensation, which often fail to accommodate object deformation and frequently introduces visual artifacts and temporal instability. To address these limitations, our approach incorporates a residual deformation framework following rigid transformation. The framework learns Gaussian deformations via adaptive scaling, opacity and color adjustments while predicting attribute residuals $(\Delta \mathbf{s}_t^l, \Delta \alpha_t^l, \Delta \mathbf{f}_t^l)$ relative to parameters of t-1 frame, ensuring both local detail preservation and temporal stability.

$$\mathbf{s}_t^l = \mathbf{s}_{t-1}^l + \Delta \mathbf{s}_t^l, \quad \alpha_t^l = \alpha_{t-1}^l + \Delta \alpha_t^l, \quad \mathbf{f}_t^l = \mathbf{f}_{t-1}^l + \Delta \mathbf{f}_t^l. \tag{7}$$

By combining both rigid transformations and residual deformations, our method effectively captures both large displacements and subtle scene variations, significantly reducing visual artifacts while maintaining temporal coherence.

**Motion-aware Adaptive Gaussian Grouping.** For long-sequence dynamic scenes with substantial motion, using only the initial frame as reference becomes inadequate due to accumulating scene variations. Meanwhile, as shown in Fig. 3(a), fixed-length group structures inevitably introduce two competing artifacts: error accumulation across frames in large groups, and data redundancy in small groups due to repeated parameter transmission. We address this through motion-aware adaptive Gaussian grouping, where the group size is dynamically determined by rigid transformation results. When the average Gaussian translation $\overline{\Delta \boldsymbol{\mu}_t}$ exceeds a predefined threshold $\tau_\mu$, indicating substantial scene changes, we initiate a new group with an updated reference frame. This adaptive grouping strategy automatically adjusts to motion intensity, employing shorter groups during rapid changes for better reference quality, while maintaining longer groups for stable segments to optimize compression efficiency. The resulting representation achieves both accuracy and compactness by balancing temporal coherence with adaptive topology updates.

In summary, our 4DGCPro dynamically structures the scene into variable-length groups for efficient temporal modeling. For a group starting at frame T with length N, we sequentially represent it as $\mathbf{G}_T, \{\Delta \boldsymbol{\mu}_t, \Delta \mathbf{R}_t, \Delta \mathbf{f}_t, \Delta \mathbf{s}_t, \Delta \alpha_t\}_{t=T+1}^{T+N-1}$, where $\mathbf{G}_T$ is the keyframe Gaussian and $\{\Delta \boldsymbol{\mu}_t, \Delta \mathbf{R}_t, \Delta \mathbf{f}_t, \Delta \mathbf{s}_t, \Delta \alpha_t\}$ are the hierarchical residual attributes. This design optimally exploits inter-frame similarities while preserving reconstruction quality under complex motions.

### 3.3 End-to-end Entropy-optimized Training

We propose an end-to-end entropy-optimized training scheme, which attains the optimal RD performance by incorporating layer-wise RD supervision and attribute-specific entropy modeling. To facilitate gradient back-propagation, we utilize differentiable quantization along with attribute-specific entropy modeling method to accurately estimate the bitrates of diverse attributes. Furthermore, we carry out progressive compression with 2D codecs on the hierarchical representation of Gaussians, enabling scalable real-time decoding and rendering. Next, we will introduce keyframe optimization, inter-keyframe optimization, and progressive bitstream generation in details.

**Keyframe Optimization.** During the optimization process of keyframes, we first use $\mathcal{L}_{color}$ as a supervision term to pretrain the Gaussians:

$$\mathcal{L}_{\text{color}} = (1 - \lambda_{\text{ssim}}) \|\mathbf{c}_g - \hat{\mathbf{c}}\|_1 + \lambda_{\text{ssim}} \mathcal{L}_{\text{D-SSIM}}, \tag{8}$$

where $\mathbf{c}_g$ and $\hat{\mathbf{c}}$ denote the ground truth and reconstructed colors respectively, and $\lambda_{\text{ssim}}$ weights the D-SSIM[36] metric. After pre-training and pruning, we hierarchically organize Gaussians and perform joint entropy-optimized hierarchical training to maximize the RD performance per level. To ensure differentiable gradient flow and enhance quantization robustness, we implement uniform noise injection $u \sim U\left(-\frac{1}{2q}, \frac{1}{2q}\right)$ to simulate quantization effects with step size $q$. Additionally, we introduce entropy estimation of Gaussian attributes into our loss function to improve compression efficiency. As shown in Fig. 3(b), keyframe Gaussian attributes exhibit irregular spatial distributions, necessitating KDE-based density estimation. Our implementation first computes the cumulative distribution function (CDF) through Silverman-rule bandwidth selection and FFT convolution, then obtains the probability density function (PDF) via numerical differentiation of the CDF:

$$P_{\text{PMF}}(\hat{y}) = P_{\text{CDF}}(\hat{y} + \frac{1}{2}) - P_{\text{CDF}}(\hat{y} - \frac{1}{2}). \tag{9}$$

To ensure optimal RD performance at each level, the keyframe optimization loss function $\mathcal{L}_{\text{key}}$ is formulated as a weighted sum of per-level losses $\mathcal{L}_{\text{key}}^l$, where each level's loss combines a photometric

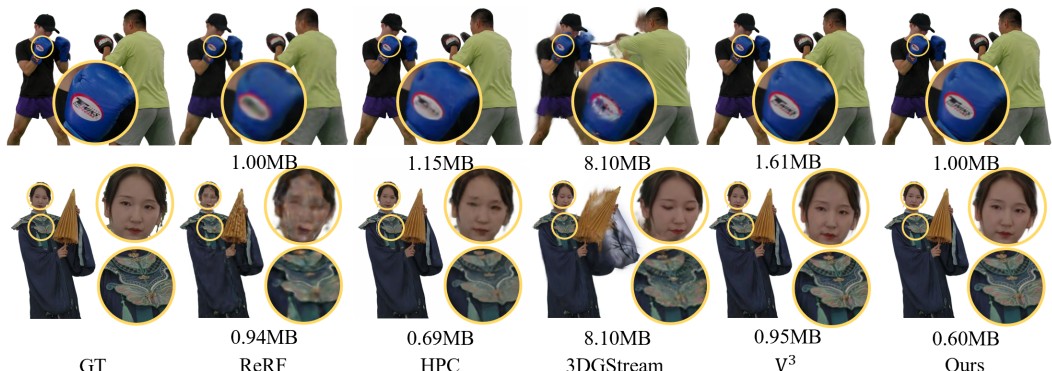

Figure 4: Qualitative comparison on our 4DGCPro and HiFi4G [26] datasets against ReRF [66], HPC [80], 3DGStream [61] and $V^3$ [69].

term $\mathcal{L}_{\text{color}}^l$ and a rate term $\mathcal{L}_{\text{rate\_key}}^l$:

$$\mathcal{L}_{\text{key}} = \sum_{l=1}^{L} \lambda^l \mathcal{L}_{\text{key}}^l = \sum_{l=1}^{L} \lambda^l (\mathcal{L}_{\text{color}}^l + \lambda_{\text{rate\_key}} \mathcal{L}_{\text{rate\_key}}^l), \tag{10}$$

$$\mathcal{L}_{\text{color}}^l = (1 - \lambda_{\text{ssim}}) \|\mathbf{c}_g - \hat{\mathbf{c}}^l\|_1 + \lambda_{\text{ssim}} \mathcal{L}_{\text{D-SSIM}}^l, \tag{11}$$

$$\mathcal{L}_{\text{rate\_key}}^l = -\frac{1}{N} \sum_{\hat{y}_t^l \in \{\hat{\mathbf{R}}_t^l, \hat{\mathbf{s}}_t^l, \hat{\mathbf{f}}_t^l, \hat{\alpha}_t^l\}} \log_2 \left( P_{\text{PMF}}(\hat{y}_t^l) \right). \tag{12}$$

Here, $\mathcal{L}_{\text{color}}^l$ measures the photometric difference between the ground truth and the Gaussian rendering results at level $l$, $\mathcal{L}_{\text{rate\_key}}^l$ denotes the entropy estimated via KDE from Gaussian attributes at the same level. $\lambda_{\text{rate\_key}}$ is the weight of the entropy loss, and $\lambda^l$ is the weight parameter for the loss at different levels. With this training objective, we obtain the keyframe Gaussians that achieve the optimal RD performance at each level.

**Inter-frame Optimization.** Building upon the hierarchically trained Gaussians of keyframes, we optimize subsequent Gaussians within each group. Since Gaussian positions and rotations are particularly crucial for rendering quality, we only employ simulated quantization deliberately exclude entropy constraints and rely solely on $\mathcal{L}_{\text{color}}$ for supervision.

In the residual deformation stage, to maximize both accuracy and compactness at each level, we maintain hierarchical supervision by augmenting color constraints with both entropy loss $\mathcal{L}_{\text{rate\_inter}}^l$ and temporal loss $\mathcal{L}_{\text{reg}}^l$. As illustrated in Fig. 3(c), we validate the Gaussian distribution of residual attributes, which allows us to simplify the entropy estimation to merely calculating the mean and variance of residuals, significantly streamlining training. To further enhance temporal coherence, we impose the temporal loss on residual attributes, explicitly enforcing inter-frame consistency. This deliberate smoothness constraint not only improves reconstruction quality but also reduces residual magnitudes during subsequent coding, ultimately optimizing storage efficiency. Thus, the training objective $\mathcal{L}_{\text{inter}}$ for this stage can be summarized as:

$$\mathcal{L}_{\text{inter}} = \sum_{l=1}^{L} \lambda^l \mathcal{L}_{\text{inter}}^l = \sum_{l=1}^{L} \lambda^l (\mathcal{L}_{\text{color}}^l + \lambda_{\text{rate\_inter}} \mathcal{L}_{\text{rate\_inter}}^l + \lambda_{\text{reg}} \mathcal{L}_{\text{reg}}^l), \tag{13}$$

$$\mathcal{L}_{\text{rate\_inter}}^l = -\frac{1}{N} \sum_{\hat{y}_t^l \in \{\Delta\hat{\mathbf{s}}_t^l, \Delta\hat{\mathbf{f}}_t^l, \Delta\hat{\alpha}_t^l\}} \log_2 \left( P_{\text{PMF}}(\hat{y}_t^l) \right), \tag{14}$$

$$\mathcal{L}_{\text{reg}}^l = \sum_{\hat{y}_t^l \in \{\Delta\hat{\mathbf{s}}_t^l, \Delta\hat{\mathbf{f}}_t^l, \Delta\hat{\alpha}_t^l\}} \|\hat{y}_t^l\|_1, \tag{15}$$

where $\mathcal{L}_{\text{inter}}^l$ denotes the inter-frame loss for the $l$-th layer of Gaussians, while $\mathcal{L}_{\text{rate\_inter}}^l$ and $\mathcal{L}_{\text{reg}}^l$ are weighted by $\lambda_{\text{rate\_inter}}$ and $\lambda_{\text{reg}}$, respectively. Through this joint entropy-optimized training framework, we obtain a compact yet high-fidelity hierarchical 4D Gaussian representation, enabling efficient volumetric video compression for storage and transmission.

**Efficient Progressive Bitstream Generation.** Once the training is completed, we explicitly separate Gaussians at different levels and implement differential quantization for Gaussian attributes, where we

Table 1: Quantitative comparison on our 4DGCPro, HiFi4G [26] and N3DV [31] datasets. Our method achieves the best rendering quality against other methods, achieving a progressive rendering results within one single model.

| Method | 4DGCPro | | | HiFi4G[26] | | | N3DV[31] | | |
|---|---|---|---|---|---|---|---|---|---|
| | PSNR (dB)↑ | SSIM↑ | Size (MB)↓ | PSNR (dB)↑ | SSIM↑ | Size (MB)↓ | PSNR (dB)↑ | SSIM↑ | Size (MB)↓ |
| ReRF[66] | 27.57 | 0.947 | 1.70 | 30.30 | 0.977 | 0.97 | 29.71 | 0.918 | 0.77 |
| HPC[80] | 27.68 | 0.948 | 1.08 | 34.14 | 0.987 | 0.72 | - | - | - |
| 3DGStream[61] | 21.08 | 0.837 | 8.1 | 21.02 | 0.946 | 8.1 | 31.54 | 0.942 | 8.10 |
| 4DGC[20] | 21.48 | 0.850 | 0.97 | 21.05 | 0.946 | 0.94 | 31.58 | 0.943 | 0.50 |
| HiCoM[16] | 24.65 | 0.926 | 2.61 | 29.37 | 0.968 | 1.94 | 31.17 | 0.939 | 0.70 |
| V³[69] | 28.11 | 0.955 | 1.60 | 36.26 | 0.994 | 0.92 | - | - | - |
| Ours(High) | **29.47** | **0.963** | 1.31 | **36.38** | **0.995** | 0.75 | **31.64** | **0.944** | 0.64 |
| Ours(Mid) | 28.68 | 0.958 | 0.66 | 35.48 | 0.991 | 0.37 | 31.14 | 0.938 | 0.43 |
| Ours(Low) | 27.69 | 0.952 | **0.33** | 34.62 | 0.988 | **0.19** | 30.68 | 0.926 | **0.21** |

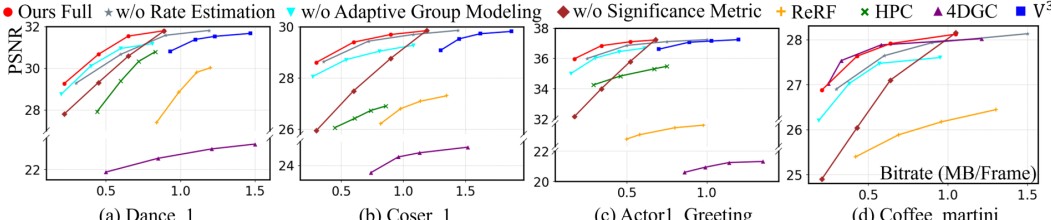

● Ours Full ⋆ w/o Rate Estimation ▼ w/o Adaptive Group Modeling ◆ w/o Significance Metric + ReRF × HPC ▲ 4DGC ■ V³

(a) Dance_1    (b) Coser_1    (c) Actor1_Greeting    (d) Coffee_martini

Figure 5: Rate-distortion curves across various datasets. Rate-distortion curves not only illustrate the superiority of our method compared to the multiple-bitrate approaches ReRF [66], HPC [80], 4DGC [20], and V³ [69], but also demonstrate the efficiency of various components within our method.

employ uint16 or uint32 precision for position information due to its higher sensitivity to errors while using uint8 for all other attributes. Since each Gaussian parameter contains multiple channels, we carefully flatten each feature channel into a separate 2D single-channel image while strictly preserving the 2D spatial continuity. The flattened feature images are systematically arranged into temporal sequences by aligning same-group, same-level, and same-channel features. These sequences are then compressed using an H.264 video encoder, enabling scalable real-time decoding and rendering via hardware video codecs and shaders. These bitstreams are transmitted collectively, enabling clients to selectively receive and decode different quality levels for adaptive rendering, thereby supporting smooth quality transitions, flexible viewing experiences, and real-time presentation.

## 4    Experiments

To comprehensively evaluate our method, we conducted experiments on two kind of distinct datasets: (1) the N3DV dataset [31] featuring subtle motions with background, and (2) the HiFi4G dataset [26] containing complex motions without background. We additionally captured a new dataset using 81 synchronized Z-CAM cinema cameras 3840×2160, recording diverse performances including dance, sports, and instrument playing. Our dataset contains not only solo performances but also multi-person interactions, which poses higher demands on the modeling method. We further introduce the detailed experimental settings in Sec. A.

### 4.1    Comparison

To validate the effectiveness of the proposed method, we compare it against several SOTA approaches including NeRF-based methods ReRF [66], HPC [80] and 3DGS-based methods 3DGStream [61], 4DGC [20], HiCoM [16], V³ [69], presenting results in Fig. 4. It can be observed that due to the limitations of the finite neural representation of NeRF when dealing with complex motions, both ReRF [66] and HPC [80] produce blurry results and over-smoothing. Meanwhile, 3DGStream [61] is limited to modeling only rigid motion of Gaussians and relies exclusively on the first frame as a universal reference across all frames. This leads to severe error accumulation over time, particularly in regions with motion, causing pronounced visual artifacts. Due to its inability to capture non-rigid deformations and significant displacements, the approach produces inconsistencies, including trajectory fragmentation and residual errors propagated from previous frames. Compared with

Table 2: The BD-PSNR results of our 4DGCPro, HPC [80], 4DGC [20] and V$^3$ [69] when compared with ReRF [66] on different datasets.

| Dataset | 4DGCPro | | | | HiFi4G[26] | | | | N3DV[31] | | | |
|---|---|---|---|---|---|---|---|---|---|---|---|---|
| Method | HPC | 4DGC | V$^3$ | Ours | HPC | 4DGC | V$^3$ | Ours | HPC | 4DGC | V$^3$ | Ours |
| BD-PSNR(dB)↑ | 3.42 | -6.15 | 1.90 | **4.20** | 5.84 | -9.10 | 7.19 | **7.87** | - | 1.99 | - | **2.07** |

Table 3: Complexity comparison of our method with dynamic scene compression methods, ReRF [66], HPC [80], 4DGC [20] and V$^3$ [69] on 4DGCPro dataset.

| Time | ReRF[66] | 4DGC[20] | V$^3$[69] | HPC[80] | | | Ours | | |
|---|---|---|---|---|---|---|---|---|---|
| | | | | High | Mid | Low | High | Mid | Low |
| Encode(ms) | 820 | 2700 | 390 | 3300 | 2870 | 2420 | 408 | 205 | **102** |
| Decode(ms) | 61 | 94 | 20 | 121 | 103 | 90 | 29 | 19 | **12** |
| Train(min) | 42.73 | **0.83** | 0.97 | 93 | 93 | 93 | 4.3 | 4.3 | 4.3 |
| Render(ms) | 52 | 5.6 | 2.8 | 231 | 167 | 110 | 3.1 | 2.5 | **2.2** |

V$^3$ [69], our method achieves better reconstruction quality while reducing the model bitrate, and it can render multi-quality reconstruction results from a single model. We further provide more demonstrations of our method in Sec. B.1.

For quantitative comparison, as demonstrated in Tab. 1, our method achieves superior performance compared to other approaches on diverse datasets. On the 4DGCPro dataset, our method achieves superior performance across all quality levels: the high-quality model attains the best PSNR (**29.47dB**) and SSIM (**0.963**) with compact size (**1.31MB**); the medium-quality version maintains high PSNR performance (**28.68dB**) with improved compression (**0.66MB**); while the low-quality configuration further reduces model size to **0.33MB** while retaining competitive quality (**27.69dB**). Notably, our framework supports multiple bitrates within a single model while outperforming baselines on all metrics. The rate-distortion superiority of our approach is further demonstrated in Fig. 5 and quantitatively validated through BD-PSNR measurements in Tab. 2. Our method achieves consistent RD improvements across all datasets and bitrates, with BD-PSNR gains of **4.20dB**, **7.87dB**, and **2.07dB** over ReRF on the 4DGCPro, HiFi4G, and N3DV datasets, respectively. These results significantly exceed those of other compared methods, including HPC (**3.42dB** on 4DGCPro) and V$^3$ (**1.90dB** on 4DGCPro). The superior RD performance demonstrates the effectiveness of our significance metric, adaptive grouping, and entropy modeling strategies.

As validated in Tab. 3, our method demonstrates exceptional computational efficiency across all quality levels. The medium-quality configuration achieves **19ms** decoding and **2.5ms** rendering per frame, enabling real-time performance at over **52 FPS**, while even the high-quality setting maintains practical efficiency with **29ms** decoding and **3.1ms** rendering. As shown in Tab. 7, our approach also delivers remarkable performance on lightweight devices: on mobile platforms, the complete pipeline requires only **43ms** for high-quality rendering, reduced to **39ms** and **34ms** for medium and low quality, demonstrating real-time decoding and rendering capability even under strict resource constraints.

These results collectively demonstrate that our method achieves the best trade-off between reconstruction quality, compression ratio, and computational efficiency among all compared approaches. The progressive coding capability further enhances practical applicability, enabling adaptive quality adjustment based on available computational resources and bandwidth constraints.

## 4.2 Ablation Studies

We conducted four ablation studies to evaluate the effectiveness of each component of our method. These experiments focus on the significance metric, motion-aware adaptive Gaussian grouping, the number of Gaussian layers, and joint entropy-optimized training. Using the full model as the baseline, we first ablated the components of the significance metric, including the weight parameter $\lambda_\Psi$. We then compared our adaptive grouping strategy against different fixed group lengths. The third experiment examines the impact of different number of Gaussian layers. Finally, we assessed various entropy modeling methods and underscored the importance of simulated quantization.

Tab. 4 presents ablation results for the significance metric and adaptive grouping strategy. The left section evaluates the significance metric for low-level Gaussians. Compared to our full model (**27.69dB**), using only opacity or volume leads to clear performance degradation (**26.71dB** and **25.83dB**, respectively). Simply multiplying these two factors also causes a significant PSNR drop of **-1.33dB**. Furthermore, improper weighting of $\lambda_\Psi$ results in measurable PSNR reductions ranging

Table 4: Ablation studies of our perceptually-weighted hierarchical Gaussian representation and adaptive Gaussian grouping.

| Significance Metric | PSNR(dB)↑ | Size(MB)↓ | | Group Size | BDBR(%)↓ | BD-PSNR(dB)↑ |
|---|---|---|---|---|---|---|
| w/o Opacity | 26.71 | 0.39 | | 1 | 48.37 | -0.96 |
| w/o Volume | 25.83 | 0.38 | | 5 | 11.81 | -0.25 |
| Multiplication | 26.36 | 0.33 | | 10 | 16.34 | -0.32 |
| $\lambda_\Psi = 2 \times 10^5$ | 27.51 | 0.34 | | 15 | 14.95 | -0.33 |
| $\lambda_\Psi = 5 \times 10^4$ | 27.57 | 0.35 | | 20 | 11.42 | -0.34 |
| Ours(Full) | **27.69** | **0.33** | | 25 | 8.11 | -0.31 |

Table 5: Ablation studies of the number of layers and end-to-end entropy-optimized training scheme.

| $L$ | BD-PSNR(dB)↑ | Training Time(min)↓ | | Training | BDBR(%)↓ | BD-PSNR(dB)↑ |
|---|---|---|---|---|---|---|
| 4 | -0.87 | 3.1 | | w/o R-E | 32.73 | -0.60 |
| 5 | -0.38 | 3.5 | | w/o H-S | 61.21 | -2.89 |
| 6 | - | 4.3 | | w/o S-Q | 4.36 | -0.1 |
| 7 | 0.06 | 4.9 | | Only KDE | 0.58 | -0.02 |
| 8 | 0.09 | 5.5 | | Only Gaussian | - | - |

from **-0.18dB** to **-0.12dB**. The right section validates our motion-aware adaptive grouping approach. Even the best-performing fixed group size strategy shows consistent degradation, with a minimum BDBR of **8.11%** and a maximum BD-PSNR reduction of **-0.25dB**, confirming the clear advantage of our adaptive grouping method in rate-distortion performance.

Table 5 examines the impact of the number of Gaussian layers and the entropy modeling strategy. The left panel shows that deeper hierarchies (e.g., $L$=8) only slightly improve BD-PSNR (up to 0.09dB) at the cost of a noticeable increase in training time (**+1.2 min**). In contrast, shallower configurations (e.g., $L$=4) lead to a more substantial degradation in BD-PSNR (**-0.87dB**). The right panel highlights the importance of entropy modeling: omitting it ("w/o R-E") introduces redundancy, while removing hierarchical supervision ("w/o H-S") severely degrades the quality of low-level Gaussians, resulting in a BD-PSNR drop of **-2.89dB**. Moreover, the absence of simulated quantization (S-Q) leads to a **4.36%** increase in BDBR, confirming its essential role in enhancing resilience to quantization errors. For Gaussian parameter modeling, using only KDE estimation ("Only KDE") achieves similar performance but prolongs training by **1.2 minutes** per frame, whereas assuming a universal Gaussian distribution ("Only Gaussian") causes training failures. Additional ablation studies are provided in Section B.3.

## 5 Discussion

**Limitation.** Although 4DGCPro presents an innovative and efficient approach to progressive streaming of volumetric video, it has several limitations. First, the Gaussian optimization process suffers from prolonged training times (several minutes) due to hierarchical supervision which leads to repeated rendering passes. Accelerating this procedure remains an essential research objective. Second, our method relies on multi-view video input and faces challenges in sparse-view reconstruction, limiting its applicability in scenarios with insufficient camera coverage. Finally, the current framework underperforms in spatially extensive scenes, necessitating further exploration to enhance its scalability.

**Conclusion.** We present 4DGCPro, a novel hierarchical 4D Gaussian compression approach for progressive volumetric video streaming. Our framework accomplishes multiple bitrate control within a sigle model while supporting both real-time decoding and high-fidelity rendering on mobile platforms, even for sequences containing large motion displacements. Our approach begins by constructing a perceptually-weighted hierarchical Gaussian representation using the importance metric. We then model inter-frame Gaussians by rigid transformations and residual deformations, enhanced by a motion-aware adaptive Gaussian grouping strategy for efficient sequence-wide modeling. Furthermore, we introduce a joint entropy-optimized training and progressive coding framework, employing attribute-specific entropy modeling to ensure precise and efficient optimization. Thanks to its multiple bitrate capability, 4DGCPro enables progressive streaming and high-efficiency decoding/rendering across multiple quality levels, making it ideal for bandwidth-fluctuating scenarios. This work establishes a critical foundation for broader volumetric video adoption.

# 6 Acknowledgements

This work is supported by National Natural Science Foundation of China (62571322, 62431015, 62271308), STCSM (24ZR1432000, 24511106902, 24511106900, 22DZ2229005), 111 plan (BP0719010), and State Key Laboratory of UHD Video and Audio Production and Presentation.

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

Figure 6: Gallery of our results. 4DGCPro delivers real-time high-fidelity rendering of scenes across challenging motions, such as "playing instruments", "dancing" and "playing sports".

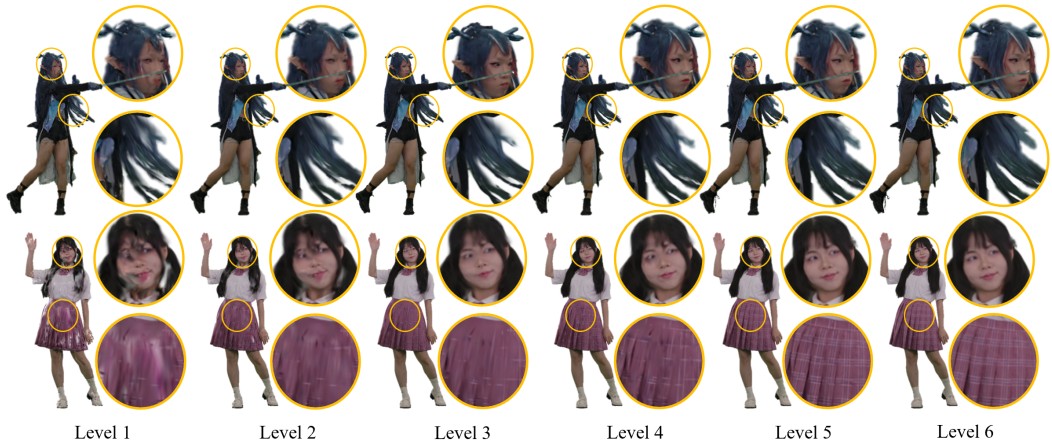

| Level 1 | Level 2 | Level 3 | Level 4 | Level 5 | Level 6 |

Figure 7: Multi-bitrate results of our method under a single bitstream.

This appendix provides additional material to supplement the main text. We will first introduce implementation details in Sec. A. Then we provide additional experimental results in Sec. B.

## A   Implementation Details

Our code is primarily based on the open-source codes of 3DGS [27] and $V^3$ [69] and is also inspired by 3DGStream [61] and 4DGC [20]. In the initialization phase, due to the limitations of the NeuS2 [70] method, it is challenging to obtain high-quality surface meshes for scenes with backgrounds, making it difficult to initialize Gaussians effectively. Therefore, on the N3DV dataset, we still initialize Gaussians based on the results of COLMAP. Additionally, we observed that in scenes with multiple interacting people, NeuS2 may occasionally fail to capture all individuals. To enhance the stability of our method, we train the NeuS2 network parameters of each keyframe to learn residuals from a pre-trained, known-correct NeuS2 network. During the pre-training phase of key frames, we first train for 12,000 steps under the setting of $\lambda_{\text{ssim}} = 0.2$. In the Gaussian pruning phase, we remove 40% of the Gaussians with lower opacity on HiFi4G dataset [26] and 4DGCPro dataset but not remove any Gaussians on N3DV dataset [31]. During the hierarchical process, we set $\lambda_\Psi = 1 \times 10^5$ to ensure a balance between volume and opacity, and divide the Gaussians into $L = 6$ layers. Regarding the motion-aware adaptive Gaussian grouping, we have selected different $\tau_\mu$ values for different

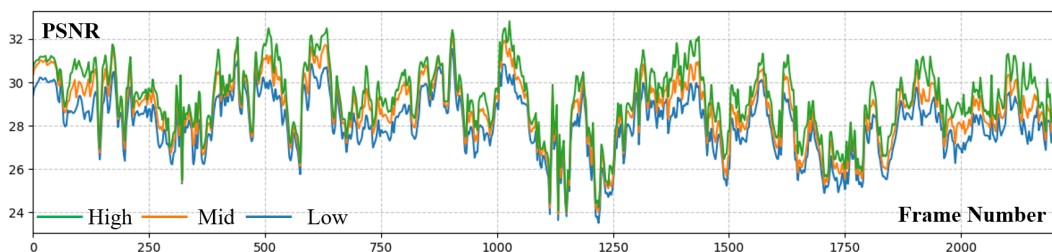

Figure 8: The results of our method on long sequences. We show that the performance of our method does not decrease as the number of frames increases.

Table 6: PSNR performance stability across three runs. Results are reported as the mean and standard deviation from three runs on the 4DGCPro dataset.

| Method | Dance1 | Dance2 | Coser1 | Coser2 | Boxing1 | Band1 | Mean |
|---|---|---|---|---|---|---|---|
| Ours(High) | 31.79 ±0.37 | 29.78 ±0.11 | 29.84 ±0.18 | 25.65 ±0.24 | 32.54 ±0.06 | 27.19 ±0.15 | 29.47 ±0.28 |
| Ours(Mid) | 30.66 ±0.28 | 29.05 ±0.14 | 29.39 ±0.13 | 25.40 ±0.21 | 30.65 ±0.08 | 26.93 ±0.10 | 28.68 ±0.22 |
| Ours(Low) | 29.26 ±0.17 | 27.97 ±0.09 | 28.60 ±0.08 | 25.06 ±0.16 | 28.99 ±0.05 | 26.23 ±0.08 | 27.69 ±0.14 |

datasets: $\tau_\mu = 0.0025$ for 4DGCpro, $\tau_\mu = 0.001$ for HiFi4G, and $\tau_\mu = 0.01$ for N3DV. Then, we conduct end-to-end entropy-optimized training on keyframes for 1,500 steps with the supervision of $\lambda^l = \begin{cases} 0.5/l, & l < L \\ 1, & l = L \end{cases}$, and $\lambda_{\text{rate\_key}} = 1 \times 10^{-7}$. In the subsequent inter-frame Optimization., we set $\lambda_{\text{rate\_inter}} = 1 \times 10^{-4}$, $\lambda_{\text{reg}} = 1 \times 10^{-3}$, and train for 800 and 2,000 steps in the rigid transformation and residual deformation phases respectively. During the compression process, we have adopted different precisions for the compression of positions depending on the dataset. Since the Gaussian positions in the N3DV dataset have a larger range, we have quantized the Gaussian positions of the N3DV dataset using uint32 precision and compressed all attributes with qp = 10. For the other two datasets, we have applied uint16 precision and qp = 20. The H.264 encoder was configured using the x264 library with the following settings: I/P-frames only (no B-frames), 3 reference frames, color space in YUV4:4:4, and preset set to "medium."

Our experimental setup includes an Intel(R) Xeon(R) W-2245 CPU running at 3.90 GHz and an RTX 3090 graphics card. In the experiment, we conducted evaluations on a total of 12 sequences from the 4DGCPro and N3DV datasets. Also, we selected the Greeting and Umbrella sequences from the HiFi4G dataset. We selected the 48th view as the test view in the HiFi4G and 4DGCPro datasets, while the 0th view was chosen in the N3DV dataset. Due to the relatively high resolution of the HiFi4G and 4DGCPro datasets, we downsample them by a factor of 2 for training. In the experimental results, we reproduce and run the comparison methods on the HiFi4G and 4DGCPro datasets, and the results on the N3DV dataset are reported in the papers of 4DGC and HiCoM [16]. Additionally, since the HPC [80] and V$^3$ methods cannot operate properly on datasets with backgrounds, we do not report their metrics on the N3DV dataset.

## B    Additional Experimental Results

### B.1    Addition Demonstrations of 4DGCPro

In this section, we additionally present some results of our method to comprehensively demonstrate the advantages of our method as much as possible. First of all, in Fig. 6, the render gallery of our method is shown. It can be seen that our method can achieve high-quality reconstruction for scenarios

Table 7: Runtime analysis on multiple platforms of rendering.

| Platform | Desktop | | | Tablet | | | Phone | | |
|---|---|---|---|---|---|---|---|---|---|
| | High | Mid | Low | High | Mid | Low | High | Mid | Low |
| Decoding(ms) | 24 | 17 | 11 | 27 | 22 | 10 | 35 | 27 | 19 |
| Rendering(ms) | 2.6 | 2.3 | 2.1 | 16 | 14 | 11 | 43 | 39 | 34 |

Table 8: Quantitative comparison of average PSNR values(dB) and model size(MB) across all sequences in the 4DGCPro dataset.

| Method | Dance1 | Dance2 | Coser1 | Coser2 | Boxing1 | Band1 | Mean |
|---|---|---|---|---|---|---|---|
| ReRF | 30.01/0.98 | 26.30/2.17 | 27.30/1.32 | 24.35/1.99 | 31.12/1.46 | 26.35/2.10 | 27.57/1.70 |
| HPC | 30.78/0.83 | 26.95/0.92 | 26.90/0.86 | 24.01/1.29 | 31.00/1.15 | 26.45/1.42 | 27.68/1.08 |
| 3DGStream | 22.65/8.10 | 18.45/8.10 | 23.62/8.10 | 19.64/8.10 | 23.49/8.10 | 18.62/8.10 | 21.08/8.10 |
| 4DGC | 22.53/0.85 | 19.77/1.00 | 24.28/0.96 | 19.06/0.94 | 24.06/0.84 | 19.18/1.20 | 21.48/0.97 |
| HiCoM | 25.06/1.62 | 23.81/1.72 | 24.55/2.07 | 21.51/3.87 | 28.87/2.92 | 24.07/3.47 | 24.65/2.61 |
| $V^3$ | 31.37/1.10 | 29.19/1.11 | 29.52/1.45 | 18.97/1.85 | **32.58**/1.61 | 27.11/2.46 | 28.11/1.60 |
| Ours(High) | **31.79**/0.89 | **29.78**/0.78 | **29.84**/1.19 | **25.65**/1.80 | 32.54/1.33 | **27.19**/1.88 | **29.47**/1.31 |
| Ours(Mid) | 30.66/0.45 | 29.05/0.39 | 29.39/0.60 | 25.40/0.91 | 30.65/0.66 | 26.93/0.94 | 28.68/0.66 |
| Ours(Low) | 29.26/**0.22** | 27.97/**0.21** | 28.60/0.30 | 25.06/**0.45** | 28.99/**0.33** | 26.23/**0.47** | 27.69/**0.33** |

Table 9: Quantitative comparison across two sequences in the HiFi4G dataset.

| Method | Greeting | | Umbrella | | Mean | |
|---|---|---|---|---|---|---|
| | PSNR (dB)$\uparrow$ | Size (MB)$\downarrow$ | PSNR (dB)$\uparrow$ | Size (MB)$\downarrow$ | PSNR (dB)$\uparrow$ | Size (MB)$\downarrow$ |
| ReRF | 29.90 | 0.96 | 30.69 | 0.98 | 30.30 | 0.97 |
| HPC | 35.47 | 0.75 | 32.81 | 0.69 | 34.14 | 0.72 |
| 3DGStream | 21.38 | 8.10 | 20.65 | 8.10 | 21.02 | 8.10 |
| 4DGC | 20.94 | 0.99 | 21.15 | 0.89 | 21.05 | 0.94 |
| HiCoM | 29.12 | 1.70 | 29.61 | 2.18 | 29.37 | 1.94 |
| $V^3$ | 37.06 | 0.89 | 35.45 | 0.95 | 36.26 | 0.92 |
| Ours(High) | **37.21** | 0.68 | **35.52** | 0.81 | **36.38** | 0.75 |
| Ours(Middle) | 36.83 | 0.34 | 34.13 | 0.39 | 35.48 | 0.37 |
| Ours(Low) | 35.96 | **0.17** | 33.28 | **0.20** | 34.62 | **0.19** |

with many complex motions. Fig. 7 shows the multi-bitrate results of our method under a single bitstream. From the figure, it can be observed that due to the high effectiveness of our hierarchical representation and layerwise supervision, 4DGCPro maintains excellent rendering quality at each level of the results. Fig. 8 presents the multi-bitrate results of our method in a long sequence. Our method can maintain a very high reconstruction quality in long sequences. As illustrated in Fig. 9, we present supplementary qualitative results for scenes from the N3DV [31] dataset. These visual outcomes vividly showcase the resilience of our approach in accurately capturing and effectively representing dynamic scenes. In addition, to verify the stability of the method, we conducted multiple experiments under the same setting and reported the fluctuations of the results, as shown in Tab. 6.

We also conducted an efficiency analysis of our 4DGCPro across multiple platforms. The test platforms consist of an Ubuntu desktop PC featuring an Intel I9-10920X processor and an NVIDIA GeForce RTX 3090 GPU, an Apple iPad powered by an Apple M2 processor, and an Apple iPhone equipped with an A15 Bionic processor. As presented in Tab. 7, we detail the time consumption of each thread within the rendering pipeline. During the decoding process, on the desktop, multi - threaded decoding combined with CUDA memory copying takes between **11ms** and **24ms**. For Apple's mobile devices, leveraging parallel decoding via compute shaders, these devices achieve a decoding time consumption comparable to that of the desktop (**22ms** on the tablet and **27ms** on the phone). Regarding the rendering thread, the desktop with a CUDA-enabled device demonstrates an extremely rapid rendering speed (**2.3ms**), while mobile devices are also capable of rendering at a satisfactory frame rate (**14ms** and **39ms**).

## B.2 Additional Comparison Results

We present the quantitative results for each scene from three datasets respectively in Tab. 8, 9 and 10 to provide a more detailed comparison. We categorize the datasets into two groups: large-motion, background-free datasets such as 4DGCPro and HiFi4G [26], and small-motion, background-containing datasets like N3DV [31]. Our method demonstrates a distinct advantage in the former group, which can be attributed to its precise motion modeling capabilities. In contrast, methods like 3DGStream [61] and 4DGC [20] perform poorly. Specifically, they completely loses its modeling ability after processing only a very short sequence due to the error accumulation. This shortcoming stems from its inherent limitations of using only the first frame as a reference and being capable of modeling only rigid motion. Notably, $V^3$ [69] shows subpar performance on the coser2 sequence.

Table 10: Quantitative comparison of average PSNR values(dB) and model size(MB) across all sequences in the N3DV dataset.

| Method | Coffee Martini | Cook Spinach | Cut Beef | Flame Salmon | Flame Steak | Sear Steak | Mean |
|---|---|---|---|---|---|---|---|
| ReRF | 26.24/0.79 | 31.23/0.84 | 31.82/0.81 | 26.80/0.78 | 32.08/0.91 | 30.03/0.51 | 29.71/0.77 |
| 3DGStream | 27.96/8.00 | 32.88/8.05 | 32.99/8.19 | 28.52/8.07 | 33.41/8.19 | 33.58/8.16 | 31.54/8.11 |
| HiCoM | 28.04/0.80 | 32.45/0.60 | 32.72/0.60 | 28.37/0.90 | 32.87/0.60 | 32.57/0.60 | 31.17/0.70 |
| 4DGC | **27.98**/0.58 | 32.81/0.44 | 33.03/0.47 | 28.49/0.51 | 33.58/0.44 | 33.60/0.50 | 31.58/0.49 |
| Ours(High) | 27.91/0.64 | **32.93**/0.67 | **33.10**/0.61 | **28.73**/0.61 | **33.77**/0.65 | **33.72**/0.66 | **31.64**/0.64 |
| Ours(Mid) | 27.63/0.43 | 32.30/0.45 | 32.51/0.41 | 28.28/0.43 | 33.09/0.43 | 33.03/0.44 | 31.14/0.43 |
| Ours(Low) | 26.88/**0.21** | 31.95/**0.22** | 32.17/**0.20** | 27.95/**0.21** | 32.49/**0.20** | 32.64/**0.23** | 30.68/**0.21** |

Table 11: Results of more ablation studies.

| Ablation Studies | High | | Mid | | Low | |
|---|---|---|---|---|---|---|
| | PSNR(dB) | Size(MB) | PSNR(dB) | Size(MB) | PSNR(dB) | Size(MB) |
| w/o Motion Decomposition | 28.84 | 1.31 | 28.17 | 0.66 | 27.04 | 0.33 |
| w/o Layer-wise Supervision | 29.53 | 1.33 | 26.49 | 0.67 | 24.98 | 0.34 |
| Ours Full | **29.47** | 1.31 | **28.68** | 0.66 | **27.69** | 0.33 |

The reason lies in the fact that NeuS2 [70] fails to initialize the two interacting objects correctly during the training process, resulting in only a portion of the image being successfully modeled. Our approach addresses this issue by introducing residual NeuS2. Within the N3DV dataset, given the small motion amplitudes, 3DGStream and 4DGC achieve good results. Nevertheless, our method still manages to deliver comparable performance.

## B.3   Additional Ablation Studies

In this section, we conducted additional experiments to validate the efficacy of motion decomposition and layer-wise supervision, as shown in Tab. 11. The results reveal that motion decomposition enables precise modeling of dynamic scenes by distinctly separating the processes of rigid transformation and residual deformation. However, simultaneously training these two components makes the positions of Gaussians less accurate and increases the training difficulty, and accurate results cannot be obtained with the same number of training steps. Regarding layer-wise supervision, while it has a marginal impact on the reconstruction quality of complete Gaussians, it yields a substantial boost in the quality of lower-level Gaussians.

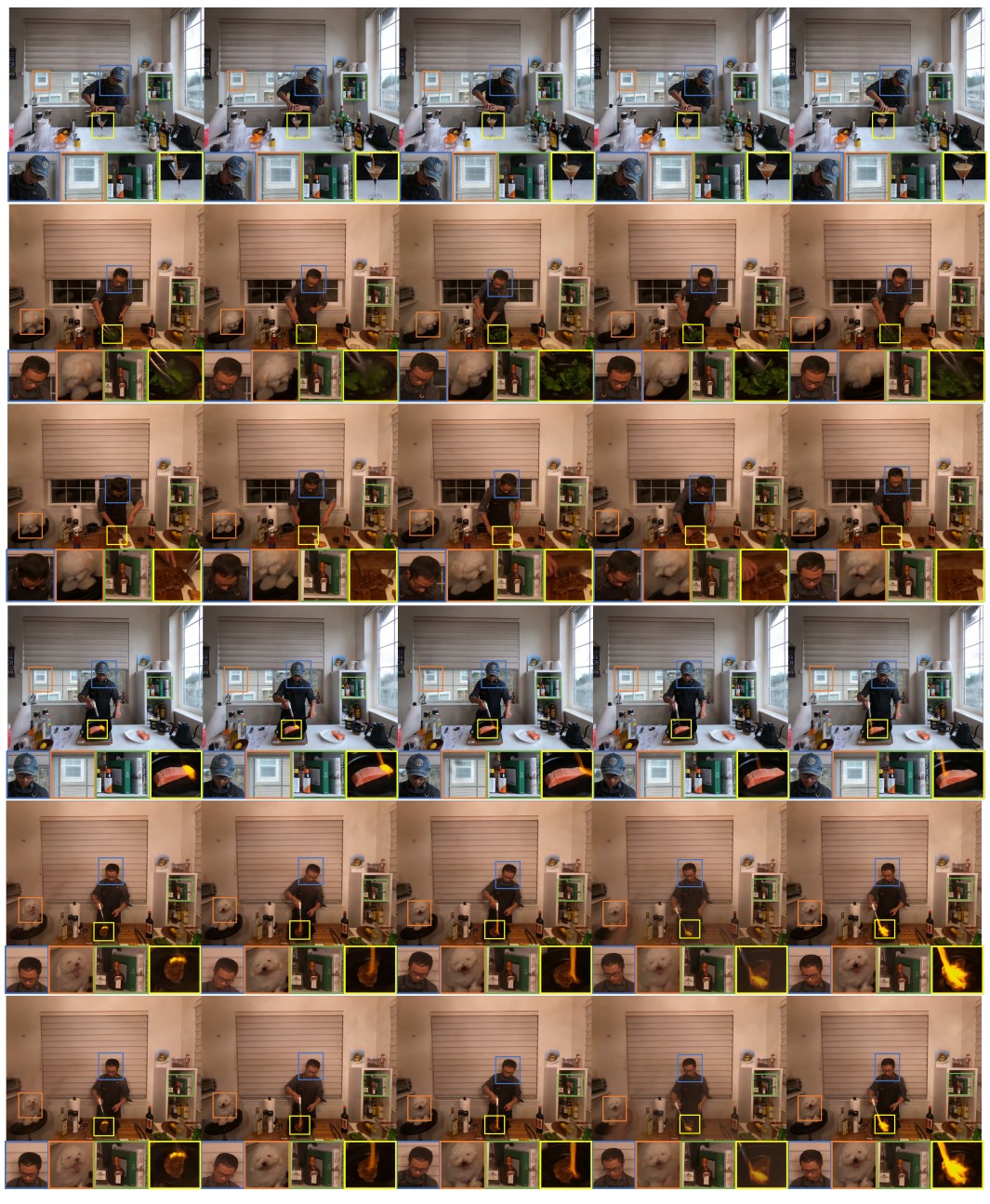

Figure 9: Qualitative results of scenes from N3DV dataset. Frames shown are the $50^{\text{th}}$, $100^{\text{th}}$, $150^{\text{th}}$, $200^{\text{th}}$, and $250^{\text{th}}$ from the test video.

