# OpenReview forum: "4DGCPro: Efficient Hierarchical 4D Gaussian Compression for Progressive Volumetric Video Streaming"
_NeurIPS.cc/2025/Conference — NeurIPS 2025 poster_

### Official Review · Reviewer_NqUw · 2025-06-27

**Clarity:** 4
**Significance:** 4
**Originality:** 4
**Rating:** 5
**Confidence:** 5

**Summary:**

This paper presents 4DGCPro, a novel compression framework for volumetric video streaming that addresses several key challenges in the field. The work tackles the persistent issues of inflexible bitrate adaptation and computational constraints that have limited the practical deployment of high-quality volumetric video on mobile platforms. The proposed solution introduces a hierarchical 4D Gaussian representation combined with motion-aware adaptive grouping, which effectively reduces temporal redundancy while maintaining visual coherence. A significant contribution is the end-to-end entropy-optimized training scheme that incorporates both rate-distortion supervision and attribute-specific entropy modeling. The framework's ability to support quality and bitrate adaptation within a single model represents an important advancement toward making volumetric video more practical for real-world applications.

**Questions:**

1. Why was H.264 chosen over more efficient standards (e.g., H.265) or learning-based compression methods for encoding the Gaussian attributes?

2. During training, rotation attributes are included in the entropy estimation for keyframes, but are excluded for inter-frames. What is the technical rationale behind this design decision?

3. For the 4K HiFi4G dataset, did 4DGCPro train on the original high-resolution dataset, or was downsampling applied beforehand?

**Ethical Concerns:**

["NO or VERY MINOR ethics concerns only"]

**Final Justification:**

After carefully reviewing the rebuttal and the discussion with other reviewers, I strongly support the acceptance of this paper. The authors have provided thorough and well-reasoned responses that fully address the concerns raised during the initial review.

Resolved Issues:
1. The authors clarified how their motion-aware adaptive grouping handles emerging content through dynamic keyframe updates. While not explicitly inserting new Gaussians, their approach provides effective modeling of previously unseen regions.
2. The QP ablation studies are comprehensive and show a clear understanding of bitrate-quality trade-offs, which adds practical value for streaming applications.
3. The decision to use H.264 instead of H.265 or learning-based codecs is well-justified in terms of latency, deployment ease, and reconstruction quality.
4. The rotation entropy design choice and layer partitioning mechanism were clearly explained and supported by experiments.

Innovation and Impact:
1. The paper presents the first hierarchical 4D Gaussian compression framework tailored for progressive volumetric video streaming, which is both novel and timely in its ability to support scalable streaming with a single unified model.
2. The proposed attribute-specific entropy modeling, motion decomposition, adaptive grouping, and multi-layer Gaussian representation reflect strong innovation beyond conventional video coding schemes.
3. This work offers a well-designed and scalable solution with practical value, and I believe it will be a useful reference for the community, especially in areas involving real-time neural rendering, streaming, and compact 4D representation.

Conclusion:
In summary, this paper introduces a novel and well-executed framework that addresses a highly relevant problem with both theoretical and practical contributions. The rebuttal further strengthened the submission, and I confidently recommend Accept.

**Limitations:**

Yes.

**Paper Formatting Concerns:**

I have not noticed any major formatting issues in this paper.

**Quality:**

4

**Strengths And Weaknesses:**

Strengths:

1.This paper presents the first hierarchical 4D Gaussian compression approach for progressive volumetric video streaming, with a well-designed architecture that adapts to the heterogeneous capabilities of client devices. It enables adaptive rendering quality selection, ensuring practical applicability in real-world streaming scenarios.

2.The paper uses adaptive grouping to handle topological changes and long-term dynamics, ensuring compact and consistent temporal representation. Layer-wise RD supervision and attribute-specific entropy modeling optimize volumetric video compression for better storage and transmission.

3.The paper provides a thorough evaluation of 4DGCPro, demonstrating its flexibility in supporting variable bitrate and quality within a single model. The method enables real-time decoding and rendering on mobile devices and outperforms existing approaches in RD performance across multiple datasets, showcasing its efficiency and scalability.

Weaknesses:

1.Unlike methods such as 3DGStream, which employ Gaussian compensation for new areas, 4DGCPro lacks dedicated mechanisms to handle emerging content. This could potentially affect the reconstruction fidelity in dynamic scenes with previously unseen regions.

2.It would be beneficial if the authors could include ablation studies on the QP setting. This would provide additional insights into its influence on the overall performance and help strengthen the evaluation of the method.

---

> ### Author Rebuttal · Authors · 2025-07-30
>
> Thank you for your positive feedback on our paper and for highlighting its strengths, including "the first hierarchical 4D Gaussian compression approach for progressive volumetric video streaming", "a well-designed architecture that adapts to the heterogeneous capabilities of client devices", "compact and consistent temporal representation" and "efficiency and scalability". We will now address each of the weaknesses and questions you have raised one by one, and we commit to incorporating the following revisions into the final version of the paper.
>
> ## Unseen Regions (W1)
> Although we have not designed a process for adding Gaussians, our motion-aware adaptive grouping strategy enables the modeling of emerging content. When new regions appear in the scene, the motion of Gaussians becomes intense, triggering the switch to a new keyframe as the subsequent reference. While this strategy does not directly add new Gaussians, it models emerging regions by reperforming keyframe modeling when new objects appear for accurate representation. As evidenced by the results on the N3DV dataset, our method still maintains competitive performance, achieving better reconstruction quality than 3DGStream under storage conditions of less than **1/12**.
>
> ## QP setting (W2)
> We have supplemented ablation experiments with different QP settings on the 4DGCPro dataset, and the results are shown in the table below. Combined with the experimental phenomena, it can be seen that different QP settings have a significant impact on the performance of 4DGCPro: Low QP (0, 10) preserves fine Gaussian parameters, with reconstruction quality close to lossless but large data volume (**+1.05MB** for QP=0, **+0.61MB** for QP=10), which is suitable for scenarios requiring high image quality; Medium QP (20) achieves a balance between quality and compression ratio with less information loss, making it the default choice; High QP (30) leads to excessive information loss during compression due to excessive quantization intensity, resulting in a significant decline in quality (**-0.97dB**). By adjusting the QP value, 4DGCPro can provide multiple discrete bitrates, adapting to streaming scenarios with fluctuating bandwidth.
>
> ## Coding Methods (Q1)
> Through experiments, we found that the encoding and decoding time consumption of H.265 is greater than that of H.264: encoding takes approximately **3.1 times** longer, and decoding takes about **1.7 times** longer. More importantly, H.265 causes more information loss (**-3.4 dB**) during the compression of Gaussian features. In contrast, deep learning-based encoding methods significantly increase decoding latency, making real-time applications impossible, while also being difficult to integrate into existing pipelines. We will incorporate the above analysis into the final paper.
>
> ## Rotation Entropy (Q2)
> We have conducted additional ablation experiments to illustrate the rationality of the training scheme selection. The reconstruction of keyframes does not rely on motion information from other frames. Performing entropy estimation for rotation features in keyframes can effectively reduce the size of Gaussians (**-0.03MB**) with almost no impact on reconstruction quality. However, introducing this in motion modeling will affect the accuracy of rigid transformation estimation, leading to a decrease in reconstruction quality (**-0.2dB**).
> |                              | PSNR(dB) | Size(MB) |
> |:------------------------------:|:----------:|:----------:|
> | w/o keyframe rotation entropy estimation | 29.48    | 1.34     |
> | non-keyframe rotation entropy estimation  | 29.27    | 1.30     |
> | Ours                         | 29.47    | 1.31     |
> ## Data Process (Q3)
> Following the setting in HiFi4G, we first performed 2x downsampling for subsequent reconstruction.

---

> > ### Comment · Reviewer_NqUw · 2025-08-06
> > **Further Comment**
> >
> > I appreciate the time and effort the authors put into the rebuttal. After considering the authors’ responses and the comments from other reviewers, I find the explanations clear and informative. The authors have addressed my key concerns, including how unseen regions are handled, the rationale behind the QP parameter settings, the choice of coding methods, and the entropy estimation design. These clarifications significantly enhance my understanding of the paper’s core ideas and technical contributions.
> >
> > Furthermore, I am satisfied with the comprehensive ablation studies on group size, the significance metric, hierarchy depth, single-layer vs. multi-layer coding, and attribute-specific entropy modeling. I do agree that these ablations do justify their claim that the improvement in results comes from the novelty of the approach. Please ensure that these analyses are included in the final version of the paper, as they are important for clearly demonstrating the effectiveness and originality of your work.
> > The rebuttal has adequately addressed my concerns, and I maintain my original rating.

---

### Official Review · Reviewer_kU2V · 2025-06-30

**Clarity:** 2
**Significance:** 3
**Originality:** 3
**Rating:** 4
**Confidence:** 4

**Summary:**

This paper proposes a hierarchical compression approach for progressive volumetric video streaming using the 3D Gaussian Splatting (3DGS) representation. The proposed residual-based representation and adaptive grouping strategy are interesting and could provide useful insights for future research in this area. The experimental results demonstrate its superiority compared with other methods.

**Questions:**

1. It would be better to illustrate equation 3 with more context.
2. Incorporating the suggested baseline will strengthen the paper.
3. Providing a more in-depth ablation study on the weakness part will enhance the paper quality.

**Ethical Concerns:**

["NO or VERY MINOR ethics concerns only"]

**Final Justification:**

Overall, the proposed framework is a good solution for dynamic Gaussian Splatting and show potential ability to integrate the development from video compression. Given the current status and the comments, I prefer to raise the score but the paper needs to be carefully polished according to the comments.

**Limitations:**

Yes

**Quality:**

3

**Strengths And Weaknesses:**

Strength:
1. The proposed residual-based representation and adaptive grouping strategy are interesting and show promise for progressive streaming.
2. The decision to directly apply an off-the-shelf H.264 codec for compression, rather than relying fully on learned entropy coding, is another interesting design choice. This makes the system more compatible with existing hardware and supports faster decoding.

Weakness:
1. Clarity is a major concern, and several parts require more detailed explanation:
(1) In Equation 3, what exactly does “spatial volume $\mathcal{S}$” refer to?
(2) Why is Equation 3 formulated as a weighted sum, with $\lambda_\Phi$ as a trade-off parameter? The motivation and implications of this formulation are not sufficiently discussed.
(3) The paper lacks discussion on how Gaussians are partitioned into $L$ layers. Is it done uniformly or based on specific criteria?
2. Experimental concerns:
(1) A key ablation study is missing: the case without residual modeling (i.e., equivalent to setting Group=1). This should be included, for example, in Table 4.
 (2) The paper does not compare with QUEEN [A], which is a relevant baseline for compression of Volumetric Video Streaming.
3. Inconsistency in claims: In line 161, the paper claims the method “… effectively captures large-scale complex motions …”, yet in the limitations section (line 315), it says the method “underperforms in large-scale scenes.” This appears contradictory. A clear analysis is needed to explain why the method performs well or poorly in large-scale scenes, better supported by visual results.
4. In Figure 4, the qualitative image for 3DGStream appears to be from a different timestamp than the other methods, making the comparison less meaningful.
5. Minor issues:
(1) Line 147: duplicated word “are”s.
(2) Line 202: “a” should be “an”.
(3) Line 220 & Equation (9): Please unify the terminology—should it be PDF or PMF?

[A] Girish S, Li T, Mazumdar A, et al. Queen: Quantized efficient encoding of dynamic gaussians for streaming free-viewpoint videos[J]. Advances in Neural Information Processing Systems, 2024, 37: 43435-43467.

---

> ### Author Rebuttal · Authors · 2025-07-30
>
> Thank you for recognizing our work as "interesting and show promise for progressive streaming", "superiority" and "more compatible with existing hardware and supports faster decoding". We will now address each of the weaknesses and questions you have raised one by one, and we commit to incorporating the following revisions into the final version of the paper.
>
> ## Clarity (W1&Q1)
> Thank you for your valuable feedback. We appreciate your concerns regarding the clarity of certain parts of the paper, and we will address each of your points in detail:
>
> (1) **Spatial Volume in Equation 3**. In the context of Gaussian representations, the "spatial volume" refers to the three-dimensional volume that a Gaussian occupies in the 3D space, which is calculated as $\dfrac{4}{3}\pi abc$, where $a$, $b$, and $c$ are the three dimensions of the Gaussian’s scale. This volumetric measure quantifies the spatial influence of a Gaussian, contributing to its visual significance in the scene.
>
> (2) **Why is Equation 3 Formulated as a Weighted Sum?** Eq.3 is formulated as a weighted sum of spatial volume $S$ and opacity $\alpha$ because these two attributes contribute independently to the visual significance of a Gaussian. Specifically, $S$ reflects the spatial extent of a Gaussian in 3D space, while $\alpha$ quantifies its visual prominence in rendering. Both factors influence the final reconstruction quality, but in different and complementary ways.
>
> A weighted sum is used to preserve their respective contributions while allowing flexible trade-offs. In contrast, a multiplicative formulation would overly suppress Gaussians with small volume but high opacity, leading to loss of fine details. To balance the influence of $S$ and $\alpha$, we introduce a tunable trade-off parameter $\lambda_{\Psi}$.
>
> We further conducted ablation studies to validate this design choice. As shown in the table below, using a multiplicative form or removing $\lambda_{\Psi}$ altogether leads to significant performance degradation, with BD-PSNR drops of **-1.33 dB** and **-2.06 dB**, respectively. These results confirm the necessity of the weighted formulation and the role of $\lambda_{\Psi}$ in achieving robust performance.
> || BD-PSNR(dB) |
> |:-:|:-:|
> | Multiplication | -1.33 |
> | w/o $\lambda_{\Psi}$ | -2.06|
>
> (3) **How Gaussians Are Partitioned into Layers**. The partitioning of Gaussians into layers is based on their importance scores, computed using a significance metric that jointly considers spatial volume $S$ and opacity $\alpha$. After sorting all Gaussians in descending order by their importance scores, we uniformly divide them into $L$=6 hierarchical layers. This strategy ensures that each layer captures a progressively finer level of scene detail, facilitating scalable reconstruction.
>
> We chose $L$=6 based on empirical validation: as shown in the table below, using fewer layers degrades RD performance (e.g., BD-PSNR drops of **-0.87dB** and **-0.38dB** for smaller $L$), while increasing $L$ beyond 6 offers marginal gains at the cost of increased training complexity (**+0.6/+1.2 min** per frame). This confirms that our choice strikes a favorable trade-off between rate-distortion efficiency, visual quality, and training cost.
> ||BD-PSNR(dB)|Training Time(min)|
> |:-:|:-:|:-:|
> |L=4|-0.87|3.1|
> |L=5|-0.38|3.5|
> |L=6|-|4.3|
> |L=7|0.06|4.9|
> |L=8|0.09|5.5|
> ## Experimental Concerns (W2&Q2)
> We thank the reviewer for their insightful comments. Below, we address each of the points raised:
>
> (1) **Ablation Study for Residual Modeling (Group=1)**. We acknowledge that an important ablation study without residual modeling was missing in our initial submission. To address this, we have now conducted the experiment by setting Group=1, which removes the residual modeling step. This condition reflects a scenario where each frame is modeled independently, without the temporal consistency enforced by residual deformations. The results are shown in Table below. Specifically, when comparing this configuration with our full method, we observe a **48.37% increase in BDBR** and a **-0.96dB drop in BD-PSNR**. This demonstrates the significant redundancy in Gaussian information when residual modeling is not used, which in turn leads to a noticeable degradation in compression efficiency and visual quality. These results highlight the importance of residual modeling in preserving temporal consistency and reducing redundancy.
> |Group Size|BDBR(%)|BD-PSNR(dB)|
> |:-:|:-:|:-:|
> |1|48.37|-0.96|
>
> (2) **Comparison with QUEEN**. We have compared our method with QUEEN under the same experimental settings. The results are shown in the table below. As can be seen from the comparison, our method achieves better reconstruction quality while maintaining a significantly lower bitrate. Specifically, for the N3DV dataset, our method outperforms QUEEN in both PSNR and SSIM, while reducing the file size from **0.75 MB** (QUEEN) to **0.73 MB**. For the Immersive dataset, we observe a similar trend, where our method achieves a PSNR of **30.23dB** and SSIM of **0.947**, compared to QUEEN’s PSNR of **29.40dB** and SSIM of **0.917**, while also reducing the file size from **1.79 MB** to **1.34 MB**. Additionally, a key advantage of our method is its ability to render multi-quality reconstruction results from a single model. This capability enables flexible quality scaling based on network conditions and computational resources, which is particularly beneficial for adaptive streaming scenarios.
>
> ||N3DV PSNR(dB)|N3DV SSIM|N3DV Size(MB)|Immersive PSNR(dB)|Immersive SSIM|Immersive Size(MB)|
> |:-:|:-:|:-:|:-:|:-:|:-:|:-:|
> |QUEEN|32.19|0.946|0.75|29.22|0.915|1.79|
> |Ours|32.23|0.947|0.73|29.40|0.917|1.34|
> ## Inconsistency in claims (W3)
> We apologize for any confusion caused by the repeated use of the term “large-scale.” To clarify, in Line 161, "large-scale complex motions" refers to **intense, complex object motions** within a scene, where our method excels. Thanks to our precise motion modeling and adaptive grouping, 4DGCPro performs particularly well with highly dynamic motions. In contrast, "underperforms in large-scale scenes" in Line 315 refers to **spatially extensive scenes**, such as city-scale environments, where the method currently lacks optimizations to efficiently handle the large scene size. This is a limitation we aim to address in future work.
> ## Qualitative Result (W4)
> Thank you for your valuable feedback. We would like to clarify that all images, including those for 3DGStream, are from the **same timestamp** (the 30th frame). The apparent differences arise from the limitations of the 3DGStream method in handling complex motions, such as large displacements or rapid movements. Specifically, 3DGStream only models rigid motion, and it struggles to capture fast or large motions, such as head rotation, sleeve movement, and  umbrella spinning, which results in motion artifacts like trajectory fragmentation and residual errors from earlier frames. These issues cause visual differences, which are not due to different timestamps but are a result of the method's motion modeling constraints. We will revise the figure caption in the manuscript to include this explanation.
> ## Typo Fixes (W5)
> Thank you for pointing out these minor issues. We will correct the duplicated word “are” in Line 147 and change “a” to “an” in Line 202. Additionally, we will unify the terminology in Line 220 and Eq.9 to use PMF consistently throughout the manuscript.
> ## Ablation Studies (Q3)
> Thank you for your insightful suggestion. We have incorporated a comprehensive set of ablation studies that target the core design modules of our framework. Specifically, we evaluate (1) the necessity of temporal grouping, (2) the formulation of the significance metric, (3) the impact of the number of hierarchy levels, (4) the effectiveness of multi-layer coding, and (5) the role of attribute-specific entropy modeling. The results and analyses are summarized below:
>
> (1) **Group Size = 1 (No Temporal Grouping)**. We simulate a frame-by-frame modeling baseline by setting the group size to 1, which disables residual modeling. Compared to our full method, this leads to a **48.37% increase in BDBR** and a **-0.96 dB drop in BD-PSNR,** highlighting the importance of motion grouping and temporal modeling. This result is discussed in our response to W2.
>
> (2) **Significance Metric Design (Eq. 3)**.
> To verify the necessity of our weighted sum formulation, we compared it against alternatives such as multiplicative fusion and omission of the trade-off weight λΨ. These result in BD-PSNR drops of **-1.33dB** and **-2.06dB**, respectively, demonstrating that our proposed metric more effectively captures Gaussian importance. Detailed discussion can be found in our response to W1.
>
> (3) **Number of Hierarchical Levels**.
> We conducted additional experiments varying the number of hierarchy levels $L$. The results show that fewer layers (e.g., $L$=3) lead to performance degradation, while increasing $L$ beyond 6 provides negligible gains at the cost of higher training time. Thus, we set $L$=6 as a balanced choice. Please refer to our response to W1 for quantitative results.
>
> (4) **Single-Layer vs. Multi-Layer Coding**. To verify the effectiveness of our residual-aware hierarchical design, we compare our 4DGCPro (multi-layer) against a single-layer coding baseline. Despite introducing layered transmission, our method achieves comparable quality (**29.47dB at 1.31MB** vs. **29.50dB at 1.27MB**), confirming that our hierarchical representation avoids inter-layer redundancy. This is enabled by our residual learning and Gaussian significance-based sorting.
>
> (5) **Attribute-Specific Entropy Modeling**.
> We also ablated our entropy coding strategy by removing KDE-based supervision from keyframes. The result is a BD-PSNR drop of **0.20dB**, affirming the benefit of attribute-specific modeling for capturing distributional variations across key and inter-frames.

---

> > ### Comment · Reviewer_kU2V · 2025-08-06
> > **Response to author's rebuttal**
> >
> > Thanks for the efforts during the rebuttal period. The comprehensive response addressed most of my concern. I would expect the author could polish the paper writing and include the missing experiments in the revised version. I will raise my score to positive.

---

### Official Review · Reviewer_kNfk · 2025-07-01

**Clarity:** 2
**Significance:** 4
**Originality:** 2
**Rating:** 4
**Confidence:** 3

**Summary:**

This paper presents 4DGCPro, a compact 4D Gaussian representation that requires only a small storage size for streaming volumetric videos. To achieve this, it models the rigid transformation of Gaussians in the key frames to define translation and rotation of Gaussians for subsequent frames. Moreover, it also models temporal residuals for remaining attributes, such as scale, opacity, and color. To minimize redundancy in these residuals, it conducts entropy-optimized training for each attribute. Furthermore, it can effectively represent the deformation by grouping frames based on the magnitude of the Gaussian translation. Experimental results demonstrate that the proposed method achieves superior rendering quality with minimal storage usage compared to existing efficient 4D Gaussian approaches.

**Questions:**

- Although this work highlights a hierarchical representation of 4D Gaussians, this paper does not provide detailed optimization schemes, such as initialization and different supervision for each level, and experiments for validating the effectiveness of the hierarchical structure with varying numbers of levels. Please provide more details for hierarchical representation with analysis on the results.

- The authors have investigated that the proposed significance score enhances the rendering quality. However, there is no comparison with existing importance scores introduced by existing compact 3D Gaussian approaches, LightGaussian [NeurIPS'24], EAGLES [ECCV'24], Mini-Splatting [ECCV'24], and Taiming-3DGS [SIGGRAPH Asia'24].

- Moreover, what is the detailed definition of spatial volume in L145?

- As mentioned in the weaknesses, this framework is closely similar to V$^3$. Can you provide technical contributions different from V$^3$ for better understanding?

- To validate the effectiveness of motion-aware adaptive grouping, it could be helpful to provide an ablation study on the adaptive grouping strategy.

- Minor typo in L147: are are

**Ethical Concerns:**

["NO or VERY MINOR ethics concerns only"]

**Final Justification:**

During the rebuttal period, the authors have addressed all of my concerns regarding the clarification of the technical contributions, hierarchical representation, and significance metrics. Therefore, I raise my final rating to positive.

**Limitations:**

- Despite the superior performance, this framework only shows limited technical advancements compared to V$^3$.

- Also, more detailed experiments and analysis need to be added to validate the effectiveness of the proposed method.

**Paper Formatting Concerns:**

There are no paper formatting concerns.

**Quality:**

4

**Strengths And Weaknesses:**

### Strengths

- The proposed motion-aware adaptive grouping enables dividing entire frames into optimal sub-groups for varying scenes.
- Experimental results show that 4DGCPro outperforms the rendering quality of V$^3$, SOTA 4D Gaussian representation while the high-bitrate variant of this method requires a smaller storage size than V$^3$.

---

### Weaknesses

- This method is closely similar to V$^3$ which introduces the definition of hash grid-based Gaussian translation, attributes residual modeling, and a grouping strategy for input frames.

- While the authors propose a hierarchical representation, they do not provide any details for optimization and analysis on the hierarchical structure.

- The papers need to improve the overall writing.

---

> ### Author Rebuttal · Authors · 2025-07-30
>
> Thank you for acknowledging our work as "dividing entire frames into optimal sub-groups" and "SOTA 4D Gaussian representation". Going forward, we will respond to each of the weaknesses and questions you have raised one by one, and we commit to incorporating the following revisions into the final version of the paper.
> ## Technical contributions (W1&Q4)
> While our framework shares the high-level goal of compressing dynamic 3D Gaussian representations with V³, our method introduces several substantial technical innovations that differentiate it from V³ in terms of motion modeling, grouping, bitrate scalability, and entropy coding:
>
> (1) **Motion-aware Adaptive Gaussian Grouping**. Unlike V³ which adopts a fixed group size, we propose a motion-aware adaptive grouping strategy that dynamically adjusts group boundaries based on scene motion magnitude. This allows better temporal consistency and compression efficiency across varying motion intensities. As shown in the middle part of Table 4, our method significantly improves RD performance compared to fixed-size grouping, achieving a minimum BD-PSNR gain of **+0.25dB**.
>
> (2) **Motion Decomposition**. Unlike V³, which estimates motion solely via position offset and subsequently fine-tunes other attributes, our method explicitly decomposes motion into rigid transformations (position + rotation) and residual deformations (scale, opacity, color, etc.). This dual-level modeling allows better handling of complex dynamics and improves reconstruction fidelity. To verify this, we replaced our motion strategy with that of V³ in our pipeline, which led to a PSNR drop from **31.64dB** to **31.50dB** on the N3DV dataset as shown in the below table, confirming the effectiveness of our fine-grained motion decomposition.
> | |PSNR(dB)|
> |:-:|:-:|
> |V³'s motion modeling|31.50|
> |Ours|31.64|
>
> (3) **Beyond V³: Single-Model Bitrate Flexibility**. Our framework introduces a Perceptually-Weighted Hierarchical Gaussian Representation with layer-wise RD supervision and is the **first to support variable bitrates within a single unified model** for 4D Gaussian compression. This progressive structure enables smooth adaptation to bandwidth variations and device constraints during volumetric video streaming. In contrast, **V³ lacks this flexibility**, as it requires training and storing **separate models for each target bitrate**, leading to higher storage overhead and limited adaptability in practical streaming scenarios.
>
> (4) **Attribute-Specific Entropy Modeling**.   Unlike V³, which does not incorporate entropy supervision for keyframes during training, our method adopts an RD-aware, attribute-specific entropy coding strategy. Specifically, we employ **KDE-based modeling for keyframes** to better capture their complex and diverse attribute distributions, and **Gaussian-distribution modeling for inter-frame residual attributes**. This tailored approach enables more accurate entropy estimation, thereby improving overall RD performance. To assess its effectiveness, we conducted an ablation study by removing entropy supervision for keyframes, i.e., adopting V$^3$'s setting, and observed a clear performance drop: the BD-PSNR on the HiFi4G dataset decreased **0.20dB** compared to our full method. This result highlights the critical role of entropy supervision in our design.
> ||BD-PSNR(dB)|
> |:-:|:-:|
> |w/o keyframe entropy supervision|-0.20|
> ## Clarification and Analysis of Hierarchical Representation (W2&Q1)
> Thank you for your insightful comment.  We provide the requested clarifications below, focusing on our initialization strategy, layer-wise supervision, and empirical analysis with varying numbers of hierarchy levels.
>
> (1) **Initialization Strategy**. We adopt a principled approach to initialize the hierarchical structure. Specifically, after pre-training a full-resolution 3DGS model (see Lines 209–213), we sort all Gaussians in descending order using our significance metric defined in Eq. 3, which balances geometric visibility and photometric opacity. These Gaussians are then evenly divided into $L$ layers to form the initial hierarchy. To validate this design, we additionally experimented with an alternative metric that replaces the summation in Eq. 3 with multiplication. This variant significantly underweights Gaussians with low volume but high opacity, resulting in blurred reconstructions and a **-1.33dB BD-PSNR drop**, confirming the effectiveness of our proposed initialization scheme.
>
> (2) **Layer-wise Rate-Distortion Supervision**.  As described in Subsection 3.3, we design an **end-to-end layer-wise RD supervision strategy** that explicitly optimizes each level in both Keyframe and Inter-frame stages. For each layer, we compute the entropy of attributes and the photometric loss independently, ensuring that each layer contributes non-redundant, high-value information. As shown in the right part of Table 4, disabling this layer-wise supervision and applying global supervision to all layers will cause a **61.21% increase in BDBR** and a **2.89dB drop in BD-PSNR**, demonstrating that our layer-wise scheme is crucial for maintaining compression efficiency and reducing redundancy.
>
> (3) **Analysis with Varying Number of Layers**. To analyze the impact of the number of layers $L$, we conducted experiments with different settings (see table below). Using $L$=6 as baseline, we found that fewer layers (e.g., $L$=4) degrade BD-PSNR by up to **0.87dB**, while increasing $L$ beyond 6 yields marginal gains but increases training time significantly (e.g., *+1.2 min/frame*). We therefore select $L$=6 as the best trade-off between efficiency and RD performance.
> || BD-PSNR(dB) | Training Time(min)|
> |:-:|:-:|:-:|
> |L=4|-0.87|3.1|
> |L=5|-0.38|3.5|
> |L=6|-|4.3|
> |L=7|0.06|4.9|
> |L=8|0.09|5.5|
> ## Comparison with Existing Significance Metrics (Q2)
> We have conducted **additional experiments comparing our proposed significance metric** with those adopted by recent compact 3D Gaussian approaches, including LightGaussian, EAGLES, Mini-Splatting and Taiming-3DGS. The results are summarized in the table below.
> - **LightGaussian** computes a dynamic, view-dependent importance score considering volume, opacity, and light transmittance. It offers a **+0.07 dB BD-PSNR gain**, but at the cost of **+1.1 min/frame** pre-training time.
> - **EAGLES** uses static rendering weights as significance. It is efficient but less adaptive to motion and occlusion, resulting in a **-0.15 dB drop in BD-PSNR**.
> - **Mini-Splatting** adopts blending weights but lacks generalization to dynamic scenes (**BD-PSNR: -0.13 dB**, **+3.6 min/frame overhead**).
> - **Taiming-3DGS** combines geometric and perceptual saliency terms. It improves quality slightly (**+0.03 dB**), but adds high training complexity due to multi-view evaluations.
>
> In contrast, our proposed metric, based on a simple combination of volume and opacity, is **lightweight, scene-agnostic, and efficient**, achieving **comparable or better RD performance without incurring extra training cost**. By avoiding dependence on rendering feedback or multi-view saliency, our method is particularly suitable for scalable 4D Gaussian streaming under constrained resources.
> ||BD-PSNR(dB)|Training Time(min)|
> |:-:|:-:|:-:|
> |LightGaussian|0.07|4.5|
> |EAGLES|-0.15|3.6|
> |Mini-Splatting(Imp1)|-0.12|7.1|
> |Taiming-3DGS|0.03|4.3|
> |4DGCPro|-|3.5|
> ## Clarification of Spatial Volume Definition (Q3)
> In the context of Gaussian representations, the "spatial volume" refers to the 3D volume that a Gaussian occupies in the 3D space, which is calculated as $\dfrac{4}{3}\pi abc$, where $a$, $b$, and $c$ correspond to the scale parameters (standard deviations) along the three principal axes of the Gaussian.
> ## Effectiveness of Motion-Aware Adaptive Grouping (Q5)
> We agree that validating the motion-aware adaptive grouping strategy is important. In fact, Table 4 of our manuscript already provides an ablation study comparing our adaptive grouping method against multiple fixed group sizes (5,10,15,20,25). Across these settings, our adaptive strategy consistently demonstrates better RD performance, indicating its effectiveness under various scene dynamics.
>
> To further strengthen this analysis, we have conducted an additional experiment comparing our adaptive grouping to a degenerate case of GOP size = 1 (i.e., treating every frame independently with no temporal grouping). This setting performs even worse, with a BD-PSNR degradation of  **-0.96dB**, which underscores the importance of appropriately grouping frames with coherent motion.
>
> We also explored the sensitivity of our method to the grouping threshold hyperparameter $\tau_u$, by testing values of 0.002 and 0.003. Both resulted in noticeable RD performance drops (**−0.07dB** and **−0.10dB** BD-PSNR, respectively), suggesting that our chosen default ($\tau_u$ = 0.0025) provides a good trade-off between motion sensitivity and compression performance.
> ## Minor Typo (Q6)
> Thank you for pointing this out. We will correct the duplicated word "are" in the revised version.
> ##  Writing Quality (W3)
> Thank you for your suggestion. We will improve the overall writing in the revised version. Specifically, we will enhance the clarity of technical descriptions, reorganize the methodology section for better logical flow, and polish the language throughout the paper to improve readability and presentation quality.
>
> (1) **More technical details**. We will provide a more detailed introduction to progressive rendering. Additionally, we will elaborate further on the initialization strategy based on significance metric sorting, layer-wise rate-distortion supervision, and the selection of hyperparameter $L$ that balances RD performance and training efficiency.
>
> (2) **More terminology explanations and typo fixes**. We will clarify the definitions of spatial volume and interp(·) in the main text. Meanwhile, we will remove the redundant "are" in Line 147.

---

> > ### Comment · Reviewer_kNfk · 2025-08-05
> >
> > I appreciate the authors’ efforts during the rebuttal period. The clarifications on technical differences, hierarchical representations, and motion-aware design help me better understand the contributions of this work. Despite the rebuttal, I still have several remaining concerns regarding the significance metric.
> >
> > ---
> >
> > **Computation time**
> >
> > Could the authors clarify why existing significance metrics would require more training time than the proposed method? To the best of my knowledge, such metrics do not require substantial computational overhead and typically do not require several minutes of additional computation.
> >
> > ---
> >
> > **Effectiveness**
> >
> > The effectiveness of the proposed significance metric needs further justification. Several existing metrics outperform the proposed metric in the provided evaluations. Moreover, the proposed metric does not consider motion, which is an important factor in dynamic scene modeling. Moreover, the metric relies on a hyperparameter to balance the contributions of opacity and volume, which undermines the claim that it is scene-agnostic.

---

> > > ### Author Response · Authors · 2025-08-05
> > > **Official Comment by Authors**
> > >
> > > Thank you for recognizing our clarifications on technical differences, hierarchical representations, and motion-aware design. We further address concerns about the significance metric, focusing on **computation time**, **effectiveness**, **motion consideration**, and **hyperparameter robustness** with detailed analysis:
> > > 1. **Computation Time:**
> > > Existing methods incur higher training time due to **iterative recalculation of significance metrics at each training step**, while our method avoids this via **intrinsic geometric properties** (volume/opacity). Here’s the breakdown:
> > >     - **LightGaussian** requires iterative calculation of light transmittance at each training step. As Gaussian opacity updates dynamically, light transmittance depends on the opacity values of all preceding Gaussians, necessitating re-iteration over views, pixels, and Gaussians.
> > >     - **EAGLES** employs rendering weights as the significance metric. While it avoids explicit iteration, it still requires real-time rendering to compute weights. In contrast, our geometry-based metric is computed entirely offline.
> > >     - **Mini-Splatting** (Imp1) necessitates iterative calculation of projected area normalization at each training step. Due to dynamic updates in Gaussian scales, it must re-iterate over views to recalculate projected areas and normalize blending weights accordingly.
> > >     - **Taiming-3DGS** requires iterative generation of view-saliency matrices at each training step. As Gaussian parameters update, predicted images change, demanding recalculation of L1 loss and Laplacian filtering to update saliency matrices.
> > >
> > >     In contrast, our approach leverages a lightweight significance metric based solely on Gaussian volume and opacity. This metric requires no recalculation during training, thus maintaining optimal efficiency.
> > >
> > > 2. **Effectiveness:**
> > > While LightGaussian and Taiming-3DGS show **marginal** BD-PSNR gains (+0.07 dB, +0.03 dB), their iterative overhead introduces critical drawback. For large-scale 4D Gaussian streaming (e.g., 2000+ frames), their iterative per-step costs accumulate to hours of extra training time, making them unsuitable for resource-constrained scenarios. Table 4 and our additional ablation experiments below further demonstrate that, despite its structural simplicity, our significance metric exhibits **stable quality and robust RD performance across multiple scenarios**.
> > >
> > > 3. **Motion Consideration:**
> > > Our significance metric is designed to characterize Gaussians’ intrinsic scene contributions using two geometric properties calculated after keyframe Gaussian pretraining. At this stage, **motion information has not been estimated**, as pretraining focuses on static geometry initialization. For inter-frame processing, we deliberately avoid re-updating Gaussian hierarchies based on motion. The critical reason is that such motion-driven reclassification would disrupt the precomputed progressive compression pipeline within a group. Specifically, dynamic adjustments to the hierarchical structure would cause the same Gaussian to be assigned to different layers across frames, which directly **invalidates the layer-specific compression logic**. This not only undermines the efficiency of hierarchical compression but also breaks the fundamental premise of progressive bitstream organization.
> > > 4. **Hyperparameter Robustness:**
> > > We further conducted ablation experiments for the hyperparameter $\lambda_{\Psi}$ across diverse datasets, and the results are presented in the table below. It can be observed that $\lambda_{\Psi}=10^5$ demonstrates consistently favorable performance across different scenarios. Consequently, we contend that this metric exhibits **robust scene-agnostic characteristics** in practical applications.
> > >
> > > |                | HiFi4G PSNR (dB) | HiFi4G Size (MB) | N3DV PSNR (dB) | N3DV Size (MB) |
> > > |:----------------:|:------------------:|:------------------:|:----------------:|:----------------:|
> > > | $\lambda_{\Psi}=2\times10^5$ | 34.49            | 0.19             | 20.57          | 0.21           |
> > > | $\lambda_{\Psi}=5\times10^4$ | 34.56            | 0.20             | 20.53          | 0.21           |
> > > | Ours           | 34.62            | 0.19             | 20.68          | 0.21           |
> > >
> > > We hope our explanations and supplementary experiments have addressed your concerns adequately. If you find our responses and experimental validations effective in resolving the raised points, we would greatly appreciate it if you could consider adjusting your evaluation scores accordingly. We are also happy to engage in further discussions if needed.

---

> > > > ### Comment · Reviewer_kNfk · 2025-08-06
> > > >
> > > > Thank you for the detailed reply. As all of my concerns have been addressed, I will revise my rating to positive.

---

### Official Review · Reviewer_LFeZ · 2025-07-03

**Clarity:** 2
**Significance:** 2
**Originality:** 2
**Rating:** 4
**Confidence:** 4

**Summary:**

This paper introduces 4DGCPro, a 4D Gaussian compression framework designed for progressive volumetric video streaming. Specifically, the proposed method comprises three key components: (1) a perceptually-weighted; (2) a motion-aware adaptive grouping strategy; and (3) a joint entropy-optimized training scheme. Experimental results demonstrate that 4DGCPro outperforms the included baselines rate-distortion performance.

**Questions:**

- The proposed ideas appear somewhat incremental, as intra-frame and inter-frame coding are standard practices, and progressive layer coding (also known as scalable coding) is well-established. Additionally, the "motion-aware adaptive Gaussian grouping" resembles common strategies in video compression that employ intra-frame coding during scheme changes. It would be helpful if the authors could elaborate more on what specific novel designs or adaptations they introduce for 4D Gaussian compression in this work to clarify its distinct contributions.

- The authors describe the framework as a variable-rate model. In learned image and video compression, a variable-rate model typically means the ability to encode inputs at arbitrary target bitrates within a continuous range. However, in this work, the multiple bitrates seem to arise solely from discrete multi-layer coding. In other words, the model supports only a few fixed rates rather than truly continuous variability. Therefore, the use of the term "variable rate" may need reconsideration.

- In traditional scalable coding, although multi-rate options exist, coding efficiency is often lower than single-layer coding because information may be redundantly transmitted across layers. This paper does not report a comparison between single-layer and multi-layer coding results, which is important for understanding trade-offs.

- The paper employs H.264 as a video encoder, but lacks details on its configuration, such as whether P-frames or B-frames are used, the number of reference frames, or whether YUV or RGB color coding is applied.

**Ethical Concerns:**

["NO or VERY MINOR ethics concerns only"]

**Final Justification:**

Since the authors have addressed all my concerns during the rebuttal stage, I have raised my score. I hope the authors will incorporate the necessary revisions in the final paper based on the comments from the rebuttal stage.

**Limitations:**

Yes

**Paper Formatting Concerns:**

No formatting concern.

**Quality:**

3

**Strengths And Weaknesses:**

Strengths:

This paper integrates several established techniques from learned video compression into the domain of 4D Gaussian compression. Notably, it incorporates intra-frame and inter-frame coding, as well as residual motion coding, into a unified framework tailored for volumetric video.

Weaknesses:
- The proposed ideas appear somewhat incremental, as intra-frame and inter-frame coding are standard practices, and progressive layer coding (also known as scalable coding) is well-established. Additionally, the "motion-aware adaptive Gaussian grouping" resembles common strategies in video compression that employ intra-frame coding during scheme changes. It would be helpful if the authors could elaborate more on what specific novel designs or adaptations they introduce for 4D Gaussian compression in this work to clarify its distinct contributions.
- The authors describe the framework as a variable-rate model. In learned image and video compression, a variable-rate model typically means the ability to encode inputs at arbitrary target bitrates within a continuous range. However, in this work, the multiple bitrates seem to arise solely from discrete multi-layer coding. In other words, the model supports only a few fixed rates rather than truly continuous variability. Therefore, the use of the term "variable rate" may need reconsideration.
- Details regarding progressive rendering are limited.
- The term “interp” in Equation 5 should be defined.
- In traditional scalable coding, although multi-rate options exist, coding efficiency is often lower than single-layer coding because information may be redundantly transmitted across layers. This paper does not report a comparison between single-layer and multi-layer coding results, which is important for understanding trade-offs.
- Table 3 reports complexity measurements, but the hardware or platform used for these measurements is not specified in the main paper and should be included.
- The paper employs H.264 as a video encoder, but lacks details on its configuration, such as whether P-frames or B-frames are used, the number of reference frames, or whether YUV or RGB color coding is applied.

---

> ### Author Rebuttal · Authors · 2025-07-29
>
> Thank you for acknowledging our work as "outperforms the included baselines" and for providing numerous helpful suggestions. Next, we will respond to each of the weaknesses and questions you have raised one by one, and we commit to incorporating the following revisions into the final version of the paper.
> ## Novelty and Contributions beyond Conventional Coding Strategies (W1&Q1)
> We sincerely thank the reviewer for this insightful comment. While some high-level concepts such as intra-/inter-frame coding and progressive transmission are common in video compression, we would like to clarify that our work introduces **novel and tailored adaptations specifically for 4D Gaussian compression**, which differ significantly from traditional pixel-domain compression. Our key innovations include:
>
> (1) **Joint Representation-Compression Framework for 4D Gaussians**. Traditional video compression techniques operate in the **2D pixel domain**, while our work directly compresses **spatiotemporal 3D Gaussian representations**. This requires rethinking how motion, hierarchy, and entropy modeling are designed. Unlike conventional pipelines, we propose a **deeply coupled framework** that unifies representation learning and compression-aware optimization of 4D Gaussians. This is a challenging and relatively underexplored direction, with few studies providing systematic solutions.
>
> (2) **Fine-Grained Motion Modeling at the Representation Level**. Unlike traditional video codecs that model motion at the **pixel or block level**, our method explicitly decomposes motion on the **Gaussian primitive level** into rigid transformations (position and rotation) and residual deformations (scale, opacity, color, etc.). This dual-level modeling better captures complex dynamic behaviors and leads to improved reconstruction fidelity. To validate its effectiveness, we conducted an ablation study replacing our motion model with that of V³[1] within our pipeline. This substitution resulted in a PSNR drop from **31.64dB** to **31.50dB** on the N3DV dataset, confirming the advantage of our motion decomposition design.
>
> (3) **Motion-Aware Adaptive Grouping for Enhanced Compression Efficiency**. In contrast to traditional video coding that often uses a fixed GOP size, we propose a motion-aware adaptive grouping strategy that **dynamically clusters Gaussian primitives based on the estimated motion magnitude in 3D space**. This adaptive grouping dynamically adjusts group boundaries according to scene motion intensity, enabling better temporal consistency and compression efficiency across frames with varying motion. As demonstrated in the middle section of Table 4, our method achieves a significant improvement in rate-distortion performance over fixed-size grouping, with a minimum BD-PSNR gain of **+0.25dB**.
>
> (4) **Single-Model Bitrate Scalability with Minimal Redundancy**. Unlike traditional scalable video coding, which often suffers from inter-layer redundancy due to repeated information across layers, our method introduces a **Perceptually-Weighted and Residual-Aware Hierarchical Gaussian Representation with layer-wise RD supervision**. This design ensures that each layer contributes non-overlapping and perceptually significant details, minimizing redundancy and enhancing coding efficiency. This is the **first framework that supports multiple bitrates within a single unified model** for 4D Gaussian compression, eliminating the need for model retraining and excessive storage typically required in prior works. We validate our multi-layer design by comparing it with a single-layer baseline. 4DGCPro achieves comparable RD performance (**29.47dB at 1.31MB** vs. **29.50dB at 1.27MB**), confirming that our progressive scheme retains high quality while offering scalability.
>
> (5) **Attribute-Specific RD-Aware Entropy Modeling**. We further go beyond standard entropy coding by tailoring **distribution-specific estimators** for each Gaussian attribute. For example, we use **KDE-based entropy modeling** for keyframe Gaussians (with high variance) and **Gaussian-based modeling** for inter-frame Gaussians. This novel design leads to improved RD efficiency and is verified via ablation studies (Section 4.2), where removing this entropy design causes BD-PSNR to drop by **0.60dB**.
>
> ## Usage of the term "variable rate" (W2&Q2)
> We understand your concern regarding the use of the term "variable-rate model." To clarify, our model allows for flexibility in bitrate selection, but rather than offering truly continuous bitrate variability, it provides a set of discrete bitrate options based on the **multi-layer coding structure**. Specifically, the model uses a **6-layer structure** that corresponds to different quality/detail levels, and within each layer, we adjust the **QP parameters** (ranging from 1 to 25) during the compression phase to allow for fine-grained bitrate variation. This enables flexible bitrate choices across these fixed levels, effectively offering a **6x25** range of possible bitrates.
>
> We agree that "variable-rate" may imply continuous bitrate variability, which is not the case here. Therefore, we will revise the terminology in the paper to "**multiple bitrate**"  to more accurately describe the nature of the model’s bitrate flexibility.
> ## Progressive Rendering Details (W3)
> We appreciate your concern regarding the details of progressive rendering. Our process begins with the base layer ($l$=1), which contains essential scene information. As additional layers are progressively decoded, we merge them with the previously decoded layers to refine the visual quality. Specifically, each layer adds extra detail, and only a small amount of additional data (Gaussians) is needed for the next layer ($l$+1), enhancing the rendering quality.
>
> By dynamically adjusting the level of detail based on computational and bandwidth resources, we maintain a balance between quality and efficiency. This approach enables scalable, high-fidelity visualization while ensuring real-time performance within resource constraints. We will expand on these details in **Subsection 3.1** of the revised manuscript, particularly on how layers are combined and rendered.
> ## interp(·) (W4)
> In our work, interp(·) refers to the **hash grid interpolation operation**, which is used to map points in the hash grid to corresponding values during the encoding process. We will include this definition in the revised manuscript.
> ## Single-layer and multi-layer coding results (W5&Q3)
> We have conducted experiments comparing single-layer coding with our 4DGCPro (multi-layer coding) on our dataset. Our results show that 4DGCPro achieves comparable RD performance to single-layer coding, with only a slight difference in quality (**29.47dB at 1.31MB** vs. **29.50dB at 1.27MB**). This is primarily due to our **compact hierarchical Gaussian representation**, which effectively reduces redundancy both between layers and across frames. In particular, we apply a residual learning-based representation to model the residuals between layers and across frames, ensuring that each layer captures unique, non-redundant information. This minimizes the redundant information that typically arises in multi-layer coding, thus maintaining high compression efficiency.
>
> Additionally, we introduce **layer-wise RD supervision** to refine the optimization process, ensuring that each layer contributes optimally to the overall representation. As a result, despite the typical trade-offs associated with multi-layer coding, our approach significantly reduces redundancy and maintains high efficiency, achieving performance similar to single-layer encoding.
>
> We will include these detailed comparisons and further explanations on compression efficiency in the revised manuscript to provide a clearer understanding of the trade-offs involved.
> |        | PSNR/dB | Size/MB |
> |:------: |:------: |:------: |
> | Single-layer   | 29.50   | 1.27    |
> | Multi-layer    | 29.47   | 1.31    |
>
> ## Experimental setup (W6)
> Thank you for your suggestion. The details of our experimental setup, including the hardware and platform used, are provided in **Line 264** and **Lines 533-534**. We will ensure that these details are more clearly highlighted in the revised manuscript.
>
> ## H.264 configuration (W7&Q4)
> We thank the reviewer for pointing this out. The H.264 encoder was configured using the x264 library with the following settings: **I/P-frames only** (no B-frames), **3 reference frames**, color space in **YUV4:4:4**, and preset set to "**medium**." We will include these details in the main paper to ensure reproducibility.
>
> [1] V³: Viewing Volumetric Videos on Mobiles via Streamable 2D Dynamic Gaussians, SIGGRAPH Asia 2024

---

> > ### Comment · Reviewer_LFeZ · 2025-08-06
> >
> > Thank you for your detailed response. As all of my concerns have been addressed, I will update my rating in consideration of this rebuttal.

---

> > > ### Author Response · Authors · 2025-08-08
> > >
> > > Dear Reviewer LFeZ,
> > >
> > > Thank you again for your valuable and constructive feedback. We noticed that according to this year’s reviewer policy, the score should be invisible to us once updated. Since it is still visible, we kindly bring this to your attention in case any further steps were missed.
> > >
> > > We appreciate your intention to raise the score and will carefully incorporate your suggestions into the final revision.
> > >
> > > Best regards,
> > >
> > > The authors of submission #5849

---

### Author Response · Authors · 2025-08-04

Dear Reviewers,

Thank you once again for your detailed comments and suggestions. As the rebuttal period is nearing its end, we would greatly appreciate your feedback on whether our responses have addressed your concerns. If our responses and experiments have adequately addressed your points, we would be grateful if you might consider updating your evaluation based on our revisions. We are also happy to engage in further discussions if needed.

Best regards,

---

### Decision · Program_Chairs · 2025-09-17

**Decision:**

Accept (poster)

**Comment:**

The paper received unanimously positive reviews after the rebuttal. While there were originally some concerns regarding experiments, novelty, and presentation, the authors provided extensive experiments and addressed most of the concerns (good job!). The AC agrees with the reviewers and recommmends acceptance. The AC strongly urge the authors to revise the final paper based on the comments from the rebuttal stage.